# Perception-R1: Pioneering Perception Policy with Reinforcement Learning

**En Yu**[1,¶], **Kangheng Lin**[2,¶], **Liang Zhao**[3,¶],
**Jisheng Yin**[3], **Yana Wei**[4], **Yuang Peng**[5], **Haoran Wei**[3], **Jianjian Sun**[3],
**Chunrui Han**[3], **Zheng Ge**[3], **Xiangyu Zhang**[3], **Daxin Jiang**[3], **Jingyu Wang**[2], **Wenbing Tao**[1†]
[1]Huazhong University of Science and Technology
[2]Beijing University of Posts and Telecommunications    [3]StepFun
[4]Johns Hopkins University    [5]Tingshua University
{yuen, wenbingtao}@hust.edu.cn

## Abstract

Inspired by the success of DeepSeek-R1, we explore the potential of rule-based reinforcement learning (RL) in MLLM post-training for perception policy learning. While promising, our initial experiments reveal that incorporating a thinking process through RL does not consistently lead to performance gains across all visual perception tasks. This leads us to delve into the essential role of RL in the context of visual perception. In this work, we return to the fundamentals and explore the effects of RL on different perception tasks. We observe that the *perceptual perplexity* is a major factor in determining the effectiveness of RL. We also observe that reward design plays a crucial role in further approaching the upper limit of model perception. To leverage these findings, we propose ***Perception-R1***, a scalable RL framework using GRPO during MLLM post-training. With a standard Qwen2-VL-2B-Instruct, Perception-R1 achieves $+\mathbf{4.2}\%$ on RefCOCO+, $+\mathbf{17.9}\%$ on PixMo-Count, $+\mathbf{4.2}\%$ on PageOCR, and notably, $\mathbf{31.9}\%$ **AP** on COCO2017 `val` for the first time, establishing a strong baseline for perception policy learning. Project code is available at `https://github.com/linkangheng/PR1`.

## 1 Introduction

> "We do not see the world as it is, but as we are — or as we are conditioned to see it."

*Stephen R. Covey*

The landscape of large language model (LLM) has undergone a paradigm shift from non-reasoning foundation model, *e.g.*, GPT-4/4o [44, 19], DeepSeek-V3 [33], to strongly reasoning model, *e.g.*, OpenAI o1/o3 [45], DeepSeek-R1 [12], and Kimi-1.5 [57]. DeepSeek-R1, in particular, introduced a simple yet effective rule-based reinforcement learning (RL) approach [55], enabling emergent reasoning patterns without relying on traditional scaffolding techniques such as Monte Carlo Tree Search (MCTS) [17, 67] or Process Reward Models (PRM) [31]. This has catalyzed a new revolution in LLM post-training techniques, prompting researchers to develop more powerful reasoning language models [42, 24].

Despite these advancements, current explorations predominantly focus on the purely linguistic domain, and the unimodal nature of these reasoning models limits their ability to engage with the world in a truly perceptive way. To bridge this gap, this work takes a pioneering step in exploring

---

[†]Corresponding author, [¶] Core contribution

39th Conference on Neural Information Processing Systems (NeurIPS 2025).

the potential of *perception policy learning* within multimodal LLMs [61, 3] from lens of RL. While transferring RL techniques with reasoning processes, *i.e.*, chain-of-thought [66], from the language domain shows promise on certain visual tasks, our empirical studies reveal that this approach is not universally effective. This inevitably prompts us to reexamine the *role that RL play in visual perception tasks, and how the utilization of RL can lead to better and scalable perception policy.*

Current understanding of reinforcement learning as a post-training technique is primarily rooted in linguistic tasks [24] and language-centric multimodal tasks [10]. This work, however, posits that *perception* is a critical prerequisite for visual reasoning. We argue that *only by fully unlocking the perceptual patterns of Multimodal LLMs (MLLMs) can these models achieve complex visual reasoning.* Visual perception tasks, fundamentally distinct from natural language tasks, necessitate a revised understanding of RL in this context due to two unique properties:

- *Visual perception is grounded in the objective physical world.* It possesses definite physical truth values (*e.g.*, points, lines, bounding boxes) but lacks the semantic depth of language.

- *Visual perception tasks, such as visual grounding and counting, are often "single-step" direct predictions.* This limits the structured reasoning search space typically explored by RL.

These characteristics indicate that applying RL to visual perception will yield different properties, motivating our exploration of a *perception-first* RL cognition. This work investigates the RL post-training of MLLMs in visual perception, complementing and extending current understanding. Through extensive experimental analysis, we have identified several bitter yet valuable lessons.

- *Explicit thinking process (CoT) during RL is not necessary for current perception policy.* (§ 5.2) We observe that the model without thinking process performs better than the one with thinking process.

- *Reward design plays a pivotal role in perception policy learning.* (§ 5.3) An appropriate reward function will lead to a healthier learning curve and explore stronger perceptual patterns of MLLM.

- *Perceptual perplexity determines RL superiority over SFT.* (§ 5.2) We observe that RL can bring more significant improvement compared to SFT on more complex visual tasks, *e.g.*, object detection.

Driven by these findings, we present a simple, effective, and scalable RL framework, *i.e.*, **Perception-R1**, for efficient perception policy learning. Inspired by mainstream language reasoning models [12, 57], Perception-R1 applies rule-based RL algorithm GRPO [55] during MLLM post-training stage. With a vanilla Qwen2-VL-2B-Instruct [61], Perception-R1 achieves significant improvement on multiple visual perception benchmarks, *e.g.*, $+4.2\%$ on RefCOCO+ [40], $+17.9\%$ on PixMo-Count [13], and $+4.2\%$ F1-score on PageOCR [34]. More importantly, Perception-R1 serves as the first time to enable a pure MLLM to reach $31.9\%$ mAP on the object detection benchmark COCO2017 [32] `val`, showcasing the great potential of general foundation models to surpass expert models in mainstream visual tasks. We hope our method, results, and analysis will inspire future research on perception policy learning with RL.

## 2 Related Works

**Multimodal Foundation and Reasoning Models.** Recently, vision-language models [37, 3, 73, 70] have demonstrated remarkable capabilities in visual comprehension [64, 68] and generation [14, 48] through large-scale pretraining [2, 61] and visual instruction tuning [37, 35]. These models integrate visual modalities into a unified semantic space via visual encoders [49] and adapters [11, 37], while leveraging auto-regressive large language models [59, 1] as decoders for output generation. Despite the advancements in multimodal foundation models, their visual reasoning capabilities remain in an early developmental stage. Recent approaches [8, 39, 41] have explored reinforcement learning (RL) post-training to enhance visual reasoning. However, they primarily focus on language-centric tasks such as ambiguous reference resolution [39] and geometric problem-solving [41], while overlooking critical aspects of perception-driven reasoning. In this work, we take a pioneering step in utilizing RL for perception policy learning, aiming to bridge this gap and advance multimodal reasoning.

**Visual Perception in Multimodal Models.** Visual perception, a core concept in computer vision [21, 52, 20, 69, 29], involves interpreting and understanding sensory (visual) information from the real world. In the context of MLLMs, visual perception is crucial for integrating, comprehending, and reasoning about visual data from images or videos. Current MLLMs typically bolster their visual perception by employing advanced visual architectures [63, 64], optimized visual-language modeling

strategies [70, 68], and sophisticated post-training techniques [74]. This work explores the potential of reinforcement learning (RL) to further enhance these visual perception capabilities.

**RL-based Post-training in LLMs and MLLMs.** Reinforcement learning (RL) has emerged as a pivotal paradigm for refining LLMs through alignment with human preferences and task-specific objectives. Prominent approaches like Reinforcement Learning from Human Feedback (RLHF) [46] and Direct Preference Optimization (DPO) [50] have demonstrated remarkable success in enhancing safety, coherence, and instruction-following capabilities of LLMs [43, 47, 44] and MLLMs [74, 60]. Recently, rule-based RL techniques, represented by GRPO [55], have demonstrated the potential for large-scale RL applications. LLMs have officially entered the era of strongly reasoning models. Subsequently, MLLMs [8, 39, 41] have also quickly followed this technology. However, so far, there has been no exciting, true "Aha Moment" in the multimodal domain. This study aims to investigate the potential contributions of RL to multimodal models, focusing on visual perception.

## 3   Preliminaries

**Perception Policy Definition.** The goal of perception policy in visual-language context is enabling the model to first (***i***) extract and understand visual information from the environment [37, 68], then (***ii***) perform logical reasoning based on this understanding [73, 70] to (***iii***) accomplish specific tasks and further interact with the environment [5, 22]. In this work, we aim to empower the model to deal with a series of pure visual, *e.g.*, *counting, detection*, and visual-language, *e.g.*, *grounding, optical character recognition (OCR)*, tasks through perception policy learning.

**Group Relative Policy Optimization (GRPO [55])** is a rule-based reinforcement learning algorithm tailored for post-training LLMs. Its core idea is to use group relative rewards to optimize the policy, eliminating the need for a separate critic model [54]. Specifically, GRPO samples multiple outputs ($\mathbf{o_1} \sim \mathbf{o_g}$ in Figure 1) from the old policy for the same input, calculates the average reward of these outputs as the baseline, and uses the relative rewards to guide policy updates. The optimization objective of GRPO can be formulated as following:

$$\mathcal{J}_{\text{GRPO}}(\theta) = \mathbb{E}_{[q \sim P(Q), \{o_i\}_{i=1}^G \sim \pi_{\theta_{\text{old}}}(O|q)]}$$

$$\frac{1}{G} \sum_{i=1}^{G} \frac{1}{|o_i|} \sum_{t=1}^{|o_i|} \left\{ \min \left[ \frac{\pi_\theta^{i,t}}{\pi_{\theta_{\text{old}}}^{i,t}} \hat{A}_{i,t}, \text{clip} \left( \frac{\pi_\theta^{i,t}}{\pi_{\theta_{\text{old}}}^{i,t}}, 1 - \epsilon, 1 + \epsilon \right) \hat{A}_{i,t} \right] - \beta \mathbb{D}_{\text{KL}}[\pi_\theta \| \pi_{\text{ref}}] \right\},$$

$$\mathbb{D}_{\text{KL}} [\pi_\theta \| \pi_{\text{ref}}] = \frac{\pi_{\text{ref}} (o_{i,t}|q, o_{i,<t})}{\pi_\theta (o_{i,t}|q, o_{i,<t})} - \log \frac{\pi_{\text{ref}} (o_{i,t}|q, o_{i,<t})}{\pi_\theta (o_{i,t}|q, o_{i,<t})} - 1, \qquad (1)$$

where $\epsilon$ and $\beta$ are hyper-parameters, and $\hat{A}_{i,t}$ is the advantage, computed using a group of rewards $\{r_1, r_2, \cdots, r_G\}$ corresponding to the outputs within each group. Refer to [12, 55] for more details.

## 4   Perception-R1

In a nutshell, our Perception-R1 applies the rule-based RL algorithm GRPO [55] to the post-training stage of MLLM and optimizes the reward modeling to support perception policy learning. Figure 1 illustrates the idea, more approach and implementation details introduced next.

### 4.1   Rule-based Reward Modeling

The reward function serves as the principal training signal in reinforcement learning (RL), directing the optimization process. Existing LLM methods [12, 57, 24] basically apply a highly resilient, rule-based reward system consisting of only two reward types: Format Reward and Answer Reward.

**Format Reward.** In existing LLM and MLLM, the output format is comprised of two essential components: the final output format and the intermediate reasoning process format. The reward for the final output is defined in accordance with specific task requirements and is typically encapsulated within <answer></answer> tags, whereas the reward for the intermediate reasoning process generally mandates that the reasoning steps be enclosed within <think></think> tags. Formally,

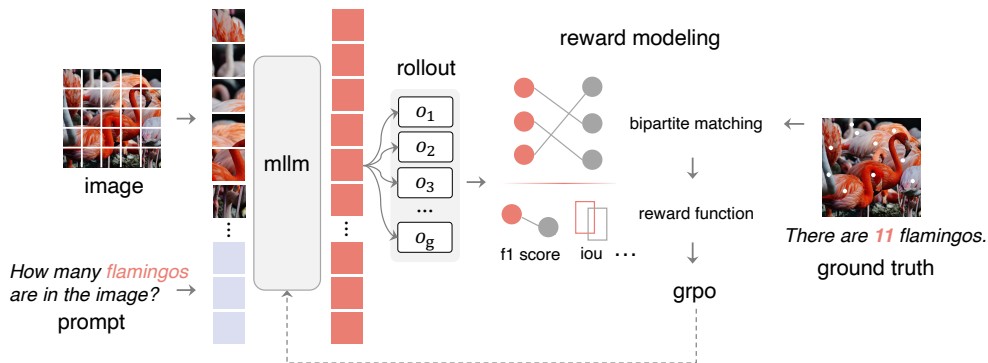

Figure 1: **Illustration of Perception-R1 framework.** Following DeepSeek-R1 [12], we prompt MLLM model to generate several rollout responses, conduct reward modeling, and then apply GRPO [55] during post-training stage.

$$S_{\text{format}} = \begin{cases} 1, & \text{if format is correct} \\ -1, & \text{if format is incorrect} \end{cases} \tag{2}$$

In Perception-R1, we follow this setting. A subtle difference emerges that visual perception task frequently require the output of object coordinates, *e.g.*, bounding box, lines, or points. Consequently, the output format must be strictly constrained to the [x1,y1,x2,y2] structure.

**Answer Reward.** The Answer Reward pertains to the correctness of model-generated responses, serving as a central consideration in reward design. Typically, outputs from language models are abstract and semantically rich, requiring validation through external mechanisms such as code-based ADE [12] or mathematical answer verification [55]. In contrast, visual perception tasks benefit from clearly defined physical ground truths, which simplify the development of a robust reward function.

Perception-R1 diverges from LLM approaches by anchoring the reward mechanism in visual discrimination. This departure is pivotal, as it replaces the often implicit and subjective feedback mechanisms typical of language models with an explicit, quantifiable metric. Formally, discriminative reward $r_i$ can be represented as:

$$r_i = \Phi(o_i, z), \tag{3}$$

where $\Phi(\cdot)$ indicates the discriminative function, for example, IoU for bounding box and euclidean distance for point. By leveraging visual discrimination, we provide the model with a clear and objective feedback signal, ensuring the model's policy update with precise measured margin.

### 4.2 Multi-Subject Reward Matching

In natural environments, physical objects rarely appear in isolation and instead frequently co-occur in groups. This inherent complexity gives rise to a challenge we define as *reward matching*, which entails aligning the model's output with the corresponding ground truth before reward computation. Specifically, when prompting the model to predict the attributes of multiple subjects within an image, *e.g.*, points and bounding box, it is necessary to determine the appropriate ground truth reference for each subject to ensure accurate reward assignment.

Formally, let $y = \{y_i\}_{i=1}^N$ denote the set of predicted attributes for $N$ subjects, and let $z = \{z_j\}_{j=1}^M$ represent the corresponding ground truth attributes. We model the reward matching problem as a bipartite graph matching task, where one set of nodes corresponds to predictions and the other to ground truths. The edge weight between a prediction $y_i$ and a ground truth $t_j$ is determined by the reward function $\Phi(y_i, z_j)$ defined in Eq. 3, which measures their similarity or compatibility. The objective is to find the optimal assignment that maximizes the total reward:

$$\hat{\sigma} = \arg\max_{\sigma \in \Omega_N} \sum_{i=1}^{N} \Phi\left(y_i, z_{\sigma(i)}\right), \tag{4}$$

where $\Omega_N$ is the set of all valid assignments between predictions and ground truths. To solve this optimization problem efficiently, we employ the Hungarian algorithm [27], a well-established method for bipartite graph matching that guarantees the optimal pairing by maximizing the overall reward (or equivalently, minimizing the cost). This ensures that each predicted attribute is accurately matched with its corresponding ground truth, thereby optimizing the reward computation process.

After the optimal reward assignment is determined, we calculate the answer reward by aggregating the individual rewards for each subject. Mathematically, the overall reward score is defined as:

$$S_{\text{answer}} = \frac{1}{N} \sum_{i=1}^{N} \Phi\left(y_i, z_{\hat{\sigma}(i)}\right),$$
$$S_{\text{total}} = S_{\text{format}} + S_{\text{answer}} \tag{5}$$

where $\hat{\sigma}$ is the optimal assignment obtained via the Hungarian algorithm. In Perception-R1, we primarily use reward matching for visual counting and object detection tasks, as these involve multiple objects.

### 4.3 Perception-R1 Configuration

**Model Setting.** Our model implementation follows Qwen2-VL [61]. We mainly use the Qwen2-VL-Instruct-2B as the baseline model. We also utilize Qwen2.5-VL-3B-Instruct [3] for training object detection tasks, due to its specialized optimization for localizing bounding boxes. The input image resolution for Qwen2-VL is dynamic cooperated with 2D-RoPE [56].

**Task and Data Setting.** Given that Perception-R1 is primarily oriented towards pure visual and visual-language tasks, we select several mainstream and representative downstream tasks for perception policy learning, specifically including *visual grounding*, *e.g.*, refCOCO [71] / + [71] / g [40], *OCR*, *i.e.*, PageOCR [34], *visual counting*, *i.e.*, Pixmo-Count [13], and *object detection*, *i.e.*, COCO2017 [32]. For each task, a subset ($5k \sim 10k$) of samples are respectively extracted as base data for individual post-training. More details are in the appendix A.1.

**Training Setting.** We focus on the RL-based post-training stage of MLLM. All the selected base models have already undergone pre-training and SFT stage. During RL stage, the initial learning rate is set as $1e-6$ with $8$ rollouts by default and a batch size of $1$. The following are some important hyper-parameters during post-training. Prompts detailed settings are in the appendix A.1.

| Gradient Accmulation | Rollout G | KL Coefficient | Max Response Len | Temperature |
|:---:|:---:|:---:|:---:|:---:|
| 2 | 8 | 0.04 | 2048 | 1.0 |

**Reward Setting.** We tailor distinct discriminative rewards for various visual perception tasks. For the grounding task, the reward is based on the Intersection over Union (IoU) between the predicted output and the ground truth. In the counting task, we adopt a paradigm similar to Qwen2.5-VL, which first detects points and then counts them. Here, the reward is derived from the Euclidean distance computed during reward matching. For OCR, the edit distance serves as the primary reward metric. Lastly, in object detection, we combine multiple rewards: an object number reward based on the F1 score, a location reward using IoU, and a binary classification reward with a missing penalty.

**Sampling Setting.** Following Kimi-1.5 [57], we adopt a curriculum sampling strategy that begins with easier data and gradually transitions to more challenging examples. Specifically, for the object detection task, we first conduct offline training on the COCO dataset to compute reward values. Based on the selected rewards, *i.e.*, number reward, we partition the dataset accordingly. As training advances, we progressively replace the data with more difficult samples (*i.e.*, those associated with lower rewards) while concurrently increasing the rollout to broaden the model's exploration space.

Table 1: **Visual grounding benchmark evaluation.** To comprehensively assess the model's grounding capability, we select referring experssion comprehension (REC) benchmark, *i.e.*, RefCOCO [71], RefCOCO+[71], and RefCOCOg[40] for evaluation. The expert model is denoted in gray.

| | | RefCOCO | | | | | | | | | | | |
|---|---|---|---|---|---|---|---|---|---|---|---|---|---|
| method | size | $val_{@50}$ | $testA_{@50}$ | $testB_{@50}$ | $val_{@75}$ | $testA_{@75}$ | $testB_{@75}$ | $val_{@95}$ | $testA_{@95}$ | $testB_{@95}$ | $val_{Avg}$ | $testA_{Avg}$ | $testB_{Avg}$ |
| MDETR [25] | - | 87.5 | 90.4 | 82.6 | - | - | - | - | - | - | - | - | - |
| OFA [62] | - | 88.4 | 90.6 | 83.3 | - | - | - | - | - | - | - | - | - |
| LLaVA-1.5 [35] | 7B | 49.1 | 54.9 | 43.3 | 10.7 | 13.6 | 6.9 | 0.4 | 0.3 | 0.3 | 20.1 | 22.9 | 16.8 |
| LLaVA-NeXT [36] | 7B | 82.5 | 88.4 | 74.0 | 45.7 | 54.8 | 35.6 | 1.9 | 2.6 | 0.7 | 43.4 | 48.6 | 36.8 |
| LLaVA-OV [28] | 7B | 73.0 | 82.3 | 63.5 | 24.2 | 29.6 | 15.9 | 0.5 | 0.5 | 0.5 | 32.6 | 37.5 | 26.6 |
| Qwen2-VL [61] | 2B | 86.8 | 89.6 | 82.0 | 77.2 | 80.6 | 70.1 | 33.0 | 35.7 | 26.9 | 65.7 | 68.6 | 59.7 |
| **Perception-R1** | 2B | **89.1** | **91.4** | **84.5** | **79.5** | **83.6** | **72.4** | **35.0** | **38.5** | **28.8** | **67.9** | **71.2** | **61.9** |

| | | RefCOCO+ | | | | | | | | | | | |
|---|---|---|---|---|---|---|---|---|---|---|---|---|---|
| method | size | $val_{@50}$ | $testA_{@50}$ | $testB_{@50}$ | $val_{@75}$ | $testA_{@75}$ | $testB_{@75}$ | $val_{@95}$ | $testA_{@95}$ | $testB_{@95}$ | $val_{Avg}$ | $testA_{Avg}$ | $testB_{Avg}$ |
| MDETR [25] | - | 81.1 | 85.5 | 72.9 | - | - | - | - | - | - | - | - | - |
| OFA [62] | - | 81.3 | 87.1 | 74.2 | - | - | - | - | - | - | - | - | - |
| LLaVA-1.5 [35] | 7B | 42.4 | 49.7 | 36.4 | 9.8 | 12.4 | 6.4 | 0.5 | 0.5 | 0.2 | 17.6 | 20.8 | 14.3 |
| LLaVA-NeXT [36] | 7B | 74.5 | 84.0 | 64.7 | 41.5 | 51.8 | 30.0 | 1.9 | 2.7 | 1.0 | 39.3 | 46.2 | 31.9 |
| LLaVA-OV [28] | 7B | 65.8 | 79.0 | 57.2 | 23.6 | 28.8 | 15.3 | 0.6 | 0.6 | 0.4 | 30.0 | 36.1 | 24.3 |
| Qwen2-VL [61] | 2B | 77.1 | 82.5 | 70.1 | 68.7 | 73.8 | 60.0 | 29.4 | 32.3 | 23.0 | 58.4 | 62.9 | 51.0 |
| **Perception-R1** | 2B | **81.7** | **86.8** | **74.3** | **73.6** | **79.3** | **64.2** | **32.6** | **36.9** | **26.7** | **62.6** | **67.7** | **55.1** |

| | | RefCOCOg | | | | | | | |
|---|---|---|---|---|---|---|---|---|---|
| method | size | $val_{@50}$ | $test_{@50}$ | $val_{@75}$ | $test_{@75}$ | $val_{@95}$ | $test_{@95}$ | $val_{Avg}$ | $test_{Avg}$ |
| MDETR [25] | - | 83.3 | 83.3 | - | - | - | - | - | - |
| OFA [62] | - | 82.2 | 82.3 | - | - | - | - | - | - |
| LLaVA-1.5 [35] | 7B | 43.2 | 45.1 | 8.5 | 9.3 | 0.3 | 0.3 | 17.3 | 18.2 |
| LLaVA-NeXT [36] | 7B | 77.5 | 77.1 | 40.7 | 39.9 | 1.8 | 1.7 | 40.0 | 39.6 |
| LLaVA-OV [28] | 7B | 70.8 | 70.8 | 23.3 | 23.6 | 0.6 | 0.7 | 31.6 | 31.7 |
| Qwen2-VL [61] | 2B | 83.3 | 83.1 | 72.7 | 73.0 | 28.9 | 27.9 | 61.6 | 61.3 |
| **Perception-R1** | 2B | **85.7** | **85.4** | **75.7** | **76.0** | **32.1** | **33.1** | **64.5** | **64.8** |

Table 2: **PageOCR evaluation,** compared with various strong expert and general models. "en" means English and "zh" means Chinese.

| | | Edit Distance ↓ | | F1-score ↑ | | Precision ↑ | | Recall ↑ | | BLEU ↑ | | METEOR ↑ | |
|---|---|---|---|---|---|---|---|---|---|---|---|---|---|
| | size | en | zh | en | zh | en | zh | en | zh | en | zh | en | zh |
| Nougat [4] | 250M | 25.5 | - | 74.5 | - | 72.0 | - | 80.9 | - | 66.5 | - | 76.1 | - |
| DocOwl1.5 [23] | 7B | 25.8 | - | 86.2 | - | 83.5 | - | 96.2 | - | 78.8 | - | 85.8 | - |
| GOT [65] | 580M | 3.5 | 3.8 | 97.2 | 98.0 | 97.1 | 98.2 | 97.3 | 97.8 | 94.7 | 87.8 | 95.8 | 93.9 |
| Qwen2-VL [61] | 2B | 8.0 | 10.0 | 94.4 | 93.0 | 96.9 | 96.1 | 93.0 | 90.5 | 90.9 | 78.0 | 94.1 | 87.2 |
| LLaVA-NeXT [36] | 7B | 43.0 | - | 64.7 | - | 57.3 | - | 88.1 | - | 47.8 | - | 58.2 | - |
| **Perception-R1** | 2B | **2.8** | **8.4** | **98.2** | **96.9** | **98.6** | **97.2** | **97.8** | **96.7** | **96.6** | **74.7** | **98.1** | **93.8** |

# 5 Experiments

The experimental section evaluates Perception-R1's performance on visual perception tasks (§ 5.1), followed by analytical experiments exploring reinforcement learning (RL)'s role in perception policy learning (§ 5.2). Finally, it discusses the interplay between visual perception and RL, along with key insights for perception policy learning (§ 5.3).

## 5.1 Performance Landscape in Perception Tasks

We evaluate Perception-R1 on mainstream perception tasks: visual grounding, counting, OCR, and object detection. Experiments use the datasets described in § 4.3 and benchmarks for image understanding. Results are in Tables 1–4.

**Visual Grounding** is a task that involves localizing visual objects based on linguistic descriptions. Specifically, given a language prompt, the model is required to output the spatial coordinates of the subject (typically a single entity) described in the prompt. As shown in Table 1, we evaluate

Table 3: **Mainstream visual tasks evaluation** including (a) visual object counting and (b) challenging general object detection. Notably, the results of expert model in (b) are copied from MMDetection [7]. † means Perception-R1 for object detection is build based on Qwen2.5-VL-3B-Instruct [3].

| | | Viusal Counting | | | | | Object Detection | | |
|---|---|---|---|---|---|---|---|---|---|
| method | size | $Pixmo_{val}$ | $Pixmo_{test}$ | method | size | epoch | AP | $AP_{50}$ | $AP_{75}$ |
| LLaVA-1.5 [35] | 7B | 33.3 | 31.0 | YOLOv3 [51] | - | 273 | 27.9 | 49.2 | 28.3 |
| LLaVA-1.6 [58] | 7B | 32.7 | 31.9 | Faster-RCNN [52] | - | 12 | 35.6 | 55.7 | 37.9 |
| LLaVA-OV [28] | 7B | 55.8 | 53.7 | DETR [6] | 41M | 500 | 42.0 | 62.4 | 44.2 |
| Qwen2-VL [61] | 2B | 60.2 | 50.5 | Qwen2.5-VL [3] | 3B | 1 | 16.1 | 23.7 | 16.7 |
| **Perception-R1** | 2B | **78.1** | **75.6** | **Perception-R1**[†] | 3B | 1 | **31.9** | **46.7** | **33.4** |

(a) **Visual counting evaluation** on Pixmo-Count [13] `val` set and `test` set.

(b) **Object detection evaluation** on COCO2017 [32] `validation` set.

Table 4: **General image understanding and reasoning evaluation,** compared with various baselines. We select 8 mainstream multimodal benchamrks, *i.e.*, MMBench [38], MMVet [72], MMStar [9], ScienceQA [53], SeedBench [18], MME [16], LLaVA-Bench [37], and ai2D [26] for the comprehensive understanding. We use the model after RL training in the counting tasks for the eval.

| | llm | MMBench | MMVet | MMStar | ScienceQA | SeedBench | MME | | LLaVA-Bench | AI2D |
|---|---|---|---|---|---|---|---|---|---|---|
| | | Avg | Avg | Avg | Avg | Avg | Cognition | Perception | Avg | Avg |
| LLaVA1.5 [35] | Vicuna1.5-7B | 62.8 | 32.8 | 32.6 | 65.4 | 60.1 | 302.1 | 1338.3 | 52.6 | 51.9 |
| LLaVA-NeXT [36] | Vicuna1.5-7B | 66.0 | 37.9 | 37.7 | 68.2 | 69.1 | 195.7 | 1419.5 | 52.7 | 67.4 |
| Qwen2-VL [61] | Qwen2-2B | **71.9** | 45.6 | **46.3** | **74.0** | 72.7 | 418.5 | 1471.1 | 46.5 | 71.6 |
| **Perception-R1** | Qwen2-2B | 71.8 | **48.9** | 45.7 | 73.4 | **73.0** | **430.0** | **1473.9** | **58.2** | **71.8** |

Perception-R1 on three mainstream benchmarks, refCOCO / + / g, and report Acc@0.5, Acc@0.75, and Acc@0.95 to comprehensively assess its visual grounding capability. We surprisingly find that several SoTA MLLMs exhibit poor performance on the more challenging Acc@0.95 metric, with scores even below 1%. In contrast, Perception-R1 achieves a stable performance of over 30% on this metric. This observation suggests that the community should prioritize reporting more discriminative results in future evaluations. The experimental results demonstrate that Perception-R1 exhibits strong competitiveness compared to both specialized and general-purpose models.

**Optical Character Recognition (OCR)** represents a critical task in visual perception due to its substantial practical value. Current methodologies predominantly adopt either expert models or fine-tuned generalist models for OCR. Perception-R1 pioneers the utilization of RL to further unlock the OCR capabilities of MLLM. As shown in Table 2, our proposed Perception-R1 achieves SoTA performance on the highly challenging OCR benchmark, *i.e.*, PageOCR [34], demonstrating significant superiority over existing expert models, *e.g.*, GOT (**98.2**% vs. **97.2**% F1-score) and robust generalist models, *e.g.*, LLaVA-NeXT (**98.2**% vs. **64.7**% F1-score). Notably, Perception-R1 does not use the Chinese OCR data for training so it is a zero-shot performance for Chinese metric. This breakthrough substantiates the formidable potential of RL applications in OCR tasks, establishing new frontiers for enhancing textual understanding and recognition in complex visual environments.

**Visual Counting**, as a fundamental vision task, necessitates models to accurately quantify category-specific instances within images, requiring robust *visual logic* to identify and enumerate targets through structured recognition patterns. In Perception-R1, we adopt a detect-then-count paradigm that reformulates the counting problem into a point detection process. As shown in Table 3a, Perception-R1 achieves remarkable counting performance, surpassing the current strong baselines by a substantial margin (**17.9**% improvement compared to Qwen2-VL in Pixmo `val` set). This advancement substantiates that RL effectively stimulates models to explore intrinsic *visual logic* mechanisms (Although counting yields deterministic results, the sequence of counting can exhibit distinct patterns), thereby enhancing their capacity to resolve complex vision tasks.

**General Object Detection**, widely regarded as the crown jewel of computer vision tasks, has long been considered one of the most challenging problems in visual perception. As a pioneering endeavor to integrate RL into object detection, Perception-R1 achieves a groundbreaking milestone, **serving as the first pure MLLM to surpass the 30+ AP threshold, *i.e.*, 31.9 AP in Table 3b, on the COCO 2017 `val` set**, matching or even exceeding the performance of specialized expert models. This achievement underscores rule-based RL's immense potential in addressing complex vision tasks requiring sophisticated visual-logic integration.

Table 5: **Ablation Study of Perception-R1.** We perform ablation studies to investigate key properties of Perception-R1 across a range of visual perception tasks. Specifically, we report the Acc@0.5 for RefCOCO / + / g val set, the F1-score for PageOCR, the average scores for Pixmo-Count, and the AP metric for COCO2017 val set. **w/o** means without. Notably, there is no reward matching applied to visual grounding and OCR tasks, as these tasks do not involve the multi-subject reward.

| case | Visual Grounding | | | OCR | Visual Counting | | Detection |
|---|---|---|---|---|---|---|---|
| | RefCOCO | RefCOCO+ | RefCOCOg | PageOCR | $Pixmo_{val}$ | $Pixmo_{test}$ | COCO2017 |
| Perception-R1 | 89.1 | 81.7 | 85.7 | 98.2 | 78.1 | 75.6 | 31.9 |
| w/o reward matching | - | - | - | - | 77.1 | 75.4 | 23.5 |
| w/o RL | 86.8 | 77.1 | 83.3 | 98.2 | 60.2 | 50.5 | 16.1 |
| w thinking | 75.1 | 67.9 | 71.3 | 93.8 | 74.9 | 72.8 | 25.7 |
| w/o thinking | **89.1** | **81.7** | **85.7** | 98.2 | **78.1** | **75.6** | 28.1 |
| RL only | **89.1** | **81.7** | **85.7** | 98.2 | **78.1** | **75.6** | **31.9** |
| SFT only | 88.2 | 80.7 | 84.6 | 97.2 | 58.0 | 59.9 | 25.9 |
| SFT+RL | 88.4 | 80.7 | 85.1 | **98.3** | 77.1 | 75.4 | 30.8 |

Table 6: **Reward design analysis of Perception-R1.** cls reward indicates binary classification reward and missing reward is a penalty to penalize missed detections. To facilitate rapid experimentation, we randomly sampled 10k data from COCO2017 train set for this experiment.

| reward function | COCO2017 | | |
|---|---|---|---|
| | AP | $AP_{50}$ | $AP_{75}$ |
| format reward | - | - | - |
| format reward + location reward (IoU) | 18.8 | 25.3 | 20.1 |
| format reward + location reward (IoU) + cls reward | 20.2 | 27.3 | 21.4 |
| format reward + location reward (IoU) + cls reward + recall reward (F1) | 27.6 | 42.0 | 28.7 |
| format reward + location reward (IoU) + cls reward + recall reward (F1) + missing reward | **28.1** | **42.0** | **29.6** |

**General Visual Comprehension** extends beyond pure perceptual tasks, and we evaluate Perception-R1 on multiple multimodal benchmarks. As shown in Table 4, we observe an intriguing phenomenon that models trained with RL for vision-specific tasks, *e.g.*, counting task, exhibit concurrent performance gains in generic comprehension benchmarks. We attribute this cross-task enhancement to the perception policy learning, which drives the model to discover superior image interpretation patterns.

## 5.2   Ablation Study of Perception-R1

In this section, we aim to conduct a comprehensive ablation study to systematically investigate the contributions of critical components within Perception-R1. Experimental results are shown in Table 5. From the experimental results, we can derive three principal empirical findings:

**Reward matching enhances the explorability of multi-subject visual perception.** As evidenced by the comparative results between row 1 and 2 in Table 5, replacing the bipartitle matching with sequential matching leads to substantial performance degradation in both visual counting and object detection task. This suggests that sequential matching constrains the RL exploration space. On the contrast, the bipartite matching mechanism provides more possibility in reward assignment, enabling the model to explore optimal visual perception patterns.

**Explicit thinking processes prove non-essential for contemporary visual perception.** Comparative analysis of row 3 and 4 reveals consistent performance degradation across all four evaluated perception tasks when incorporating an explicit thinking process during both training and inference phases. Similar phenomenon also emerges in image classification tasks [30]. We posit that this phenomenon arises because *current visual perception tasks are more oriented toward visual logic rather than semantic logic.* This shift implies that explicit language-centric reasoning processes are unnecessary, as models tend to focus more on learning ***implicit*** visual patterns.

**Perceptual perplexity dictates RL's advantage over SFT.** A comparison of post-training methods (SFT, RL, and SFT+RL) across four perception tasks (Table 5, rows 6-8) reveals that RL offers superior performance enhancement in tasks with high perceptual perplexity, such as counting and multi-object/category detection. Conversely, for low-perplexity tasks like grounding and OCR, RL performs comparably to or even worse than SFT. This suggests that high perceptual perplexity is a significant factor influencing RL's effectiveness, indicating that RL techniques are most beneficial for tasks with greater perceptual complexity and a larger exploration space for the perception policy. Further analysis of perceptual perplexity is provided in the appendix A.2.

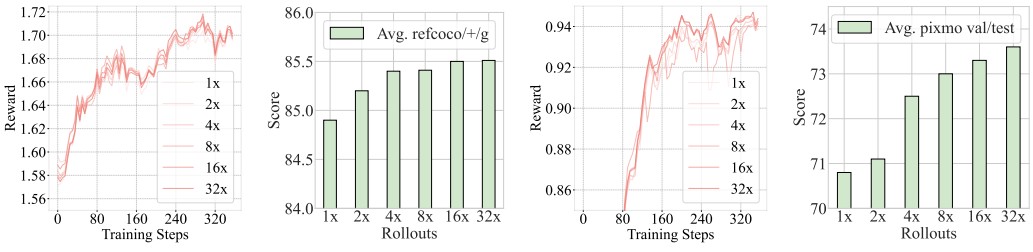

| (a) Grounding reward | (b) Grounding performance | (c) Counting reward | (d) Counting performance |

Figure 2: **Scalability analysis of Perception-R1.** We select two primary tasks: grounding and counting. We visualize the training reward curves under varying numbers of rollouts and evaluate the final performance of each task. All experiments are conducted with $5k$ sampled data. And the default rollout number setting ($1\times$) is $8$.

## 5.3 More In-depth Analysis

In this section, we explore several key properties of Perception-R1 to further enhance our understanding of Perception Policy Learning with RL.

**Analysis of reward design for perception policy learning.** We introduced the details of reward function of Perception-R1 in § 4.3. In this part, we examine the influence of these reward functions on perception policy learning. Specifically, using object detection as a case study, we incrementally integrate the designed answer reward into the format reward, as illustrated in Table 6. The results indicate that the progressive introduction of refined reward functions leads to consistent improvements in detection performance, ultimately exceeding the performance of expert models. This underscores the critical role of reward design in perception policy learning. Furthermore, it identifies a promising avenue for future research: *the development of more refined and task-specific reward functions to enhance perception policy learning.*

**Analysis of scaling up rollout for perception policy learning.** The scalability of RL is a key concern of existing LLM post-training. In this part, we analyze the scalability of Perception-R1, focusing specifically on scaling up the number of rollouts. As shown in Figure 2, we conduct rollout-scaling experiments in two tasks: visual grounding and visual counting. The results indicate that increasing rollout count enhances reward optimization and final performance. This demonstrates Perception-R1's strong scaling properties and underscores the critical role of rollout quantity in scaling perception policies. By generating sufficient rollouts, the model broadens its exploration space, increasing the diversity of candidate solutions for reward evaluation. This expansion accelerates convergence to optimal visual perception patterns.

# 6 Limitation and Conclusion

*"What can RL bring to MLLM?"* is a public question since the propose of DeepSeek-R1. Several latest works attempt to apply RL from the perspective of language-centric visual reasoning [39, 15, 41]. However, in this paper, we take a different pathway and argue that perception is a crucial prerequisite for visual reasoning. *Only by fully unlocking the perception patterns of MLLMs can the models possess the ability to reason about complex visual tasks.* Nevertheless, we regrettably find that many current perception tasks are overly simplistic, which limits the exploration space for RL. This, in turn, restricts the possibility of MLLMs achieving a perceptual "Aha moment" through thinking process. Finding more appropriate perception tasks, *aka., meta task*, may be the key to addressing this issue.

In a summary, this work takes a pioneering step in exploring the potential of rule-based RL in MLLM post-training for perception policy learning. Through extensive experimental analysis, we establish several valuable cognition about perception policy learning with RL. Driven by these findings, we build **_Perception-R1_**, a simple, effective, and scalable RL framework for efficient perception policy learning. Perception-R1 sets new SoTAs across multiple visual perception tasks, particularly in object detection tasks. By introducing a novel paradigm, it achieves and even surpasses the performance of expert models, showing the great potential of perception policy learning with RL.

## Acknowledgements

This research is supported in part by the National Key Research and Development Program of China (Grant No. 2023ZD0121300), and National Natural Science Foundation of China (Grant No. 62495092).

This research is also supported by the National Natural Science Foundation of China under Grant 62576144 and Grant 62176096.

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

# A Technical Appendices and Supplementary Material

In this appendix, we provide additional details about **Perception-R1**, which are omitted due to the 9-page limit of the main paper. Specifically, Section A.1 elaborates on the detailed dataset and training settings. Section A.2 presents additional explanation about perceptual perplexity. Section A.3 presents more experimental results.

## A.1 Additional Details about Experimental Setting

**More detailed dataset information of Perception-R1.** In Section 4.3, we introduced what data was used for RL post-training of Perception-R1 on which tasks. In this part, we will provide more detailed information about the datasets, as shown in Table 7.

Table 7: **Training dataset statistics.** Notably, we do not mix the data from different perception tasks for joint training because the rewards for different tasks vary.

| tasks | datasets | Original | Used | Ratio |
|---|---|---|---|---|
| visual grounding | RefCOCO / RefCOCO+ / RefCOCOg | 320k | 5k | 1.56% |
| OCR | PageOCR | 50k | 5k | 10% |
| visual counting | PixMo-Count | 1.9M | 10k | 0.5% |
| object detection | COCO2017 | 110k | 110k | 100% |
| overall | - | 2.38M | 130k | - |

**More detailed training setting information of Perception-R1.** Section 4.3 elaborates on several key parameters of Perception-R1. In this part, we further demonstrate the diverse prompts employed for distinct perception tasks, as shown in Table 8.

Table 8: **Prompts of Perception-R1.** The system prompt of Perception-R1 follows Qwen2-VL [61] and Qwen2.5-VL [3].

| tasks | system prompt | user prompt |
|---|---|---|
| visual grounding | Qwen2-VL | Output the bounding box of the {question} in the image. |
| OCR | Qwen2-VL | OCR this image. |
| visual counting | Qwen2-VL | Output all the bounding boxes of the {label} |
| object detection | Qwen2.5-VL | Please output bbox coordinates and names of {90 categories of COCO}. |

## A.2 Additional Explanation about Perceptual Perplexity

In this work, we point out that *perceptual perplexity* is a major factor in determining the effectiveness of RL. And RL techniques are most beneficial for tasks with greater perceptual complexity and a larger exploration space for the perception policy. In this part, we aim to demonstrate the impact of perceptual perplexity across different tasks on the performance of RL through quantitative analysis. Specifically, we utilize the theoretical possibility of matching outcomes in reward matching to quantify the perceptual perplexity of the task, that is, how many theoretical matching results exist between the model's predictions and the ground truth. We also count the probabilities of actual matches across different datasets, that is, the average number of permutations of the ground-truth output per image.

Table 9: **Perceptual perplexity analysis** of different perception tasks. $n$ is the number of ground truth. The statistical value of perplexity is calculated based on sampled data.

| tasks | dataset | theoretical | statistical |
|---|---|---|---|
| visual grounding | refCOCO | 1 | 1 |
| OCR | PageOCR | 1 | 1 |
| visual counting | Pixmo-Count | $A_n^n$ | 229,202 |
| object detection | COCO2017 | $A_n^n$ | 3.47e+83 |

As shown in Table 9, in visual grounding and OCR tasks, there is only one fixed target for the ground truth, leading to just one possible matching outcome. However, for visual counting and

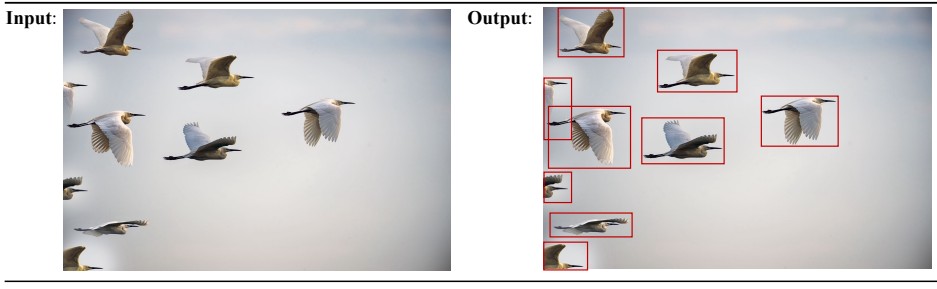

**Input:** **Output:**

**Question:**

    \<image\>Output all the bounding boxes of the bird.

**Perception-R1:**

    (259,427),(466,591)(298,172),(497,315)(558,344),(759,524)(0,273),(61,466)(30,790),(222,875)(27,9),(208,207)(0,902),
(95,998)(0,637),(51,714)(15,389),(215,599)      Num boxes = 9

Figure 3: **Demo case of Percpetion-R1** on visual counting task.

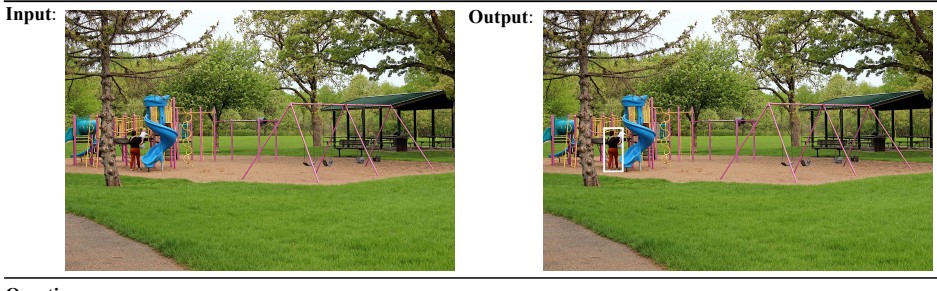

**Input:** **Output:**

**Question:**

    \<image\>Output the bounding box of the woman holding a child in the image.

**Perception-R1:**

    The woman holding a child(157,459),(208,622)

Figure 4: **Demo case of Percpetion-R1** on visual grounding task.

object detection tasks, multiple target objects exist in an image. The various permutations of these targets create numerous matching possibilities, offering ample exploration space for RL to optimize perception capabilities.

## A.3 Additional Experimental Results

**More qualitative results.** In this section, we provide more qualitative analysis of Perception-R1 on multiple visual perception tasks. The selected cases are shown in Figure 3 - 5 including visual counting, visual grounding, object detection, and OCR task, respectively.

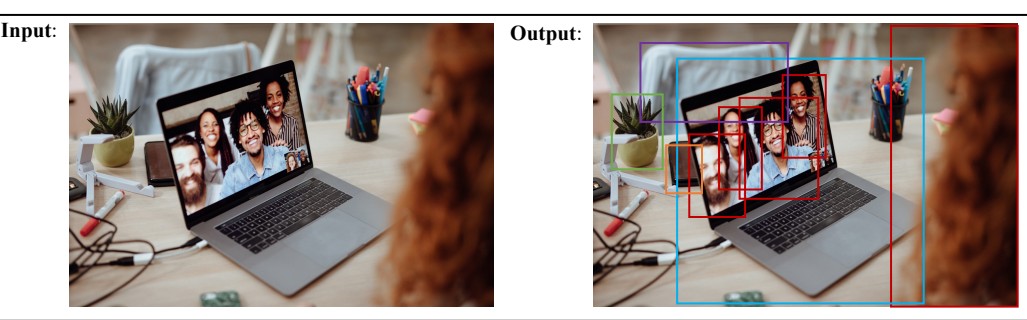

**Question**:

        <image>Please output bbox coordinates and names of {90 categories of COCO}.

**Perception-R1**:

```
[
{"bbox_2d": [435, 1, 616, 415], "label": "person"},
{"bbox_2d": [218, 112, 323, 253], "label": "person"},
{"bbox_2d": [147, 165, 225, 280], "label": "person"},
{"bbox_2d": [183, 126, 239, 242], "label": "person"},
{"bbox_2d": [276, 77, 338, 192], "label": "person"},
{"bbox_2d": [127, 54, 471, 406], "label": "laptop"},
{"bbox_2d": [108, 174, 156, 243], "label": "cell phone"},
{"bbox_2d": [26, 107, 103, 214], "label": "potted plant"},
{"bbox_2d": [64, 29, 280, 137], "label": "chair"}
]
```

Figure 5: **Demo case of Percpetion-R1** on general object detection task. The color of bounding boxes correspond to the category.

**Input**:

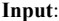

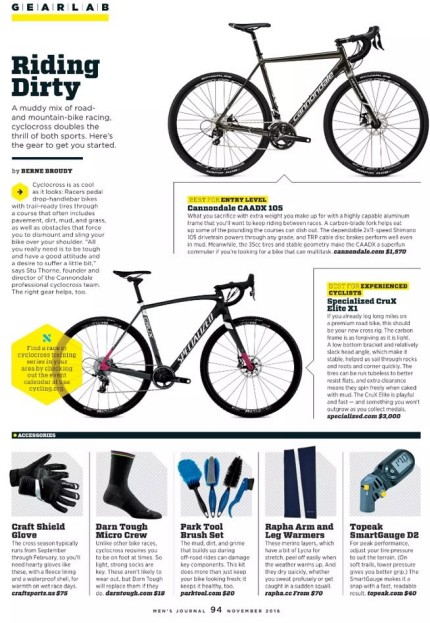

**Question**:

<image>OCR this image.

**Perception-R1**:

G E A R L A B\n\nRiding Dirty\n\nA muddy mix of road- and mountain-bike racing, cyclocross doubles the thrill of both sports. Here's the gear to get you started.\n\nby Berne Broudy\n\nCyclocross is as cool as it looks: Racers pedal drop-handlebar bikes with trail-ready tires through a course that often includes pavement, dirt, mud, and grass, as well as obstacles that force you to dismount and sling your bike over your shoulder. "All you really need is to be tough and have a good attitude and a desire to suffer a little bit," says Stu Thorne, founder and director of the Cannondale professional cyclocross team. The right gear helps, too.\n\nBest for Entry Level\n\nCannondale CAADX 105\n\nWhat you sacrifice with extra weight you make up for with a highly capable aluminum frame that'll keep you going between races. A carbon-blade fork helps eat up some of the pounding the courses can dish out. The dependable 2x11 speed Shimano 105 drivetrain powers through any grade, and TRP cable disc brakes perform well even in mud. Meanwhile, the 35cc tires and stable geometry make the CAADX a superfun commuter if you're looking for a bike that can multitask.\n\nncannondale.com $1,970\n\nBest for Experienced Cyclists\n\nSpecialized CruX Elite X1\n\nIf you already log long miles on a premium road bike, this should be your new cross rig. The carbon frame is as light as it is.\n\nA low bottom bracket and relatively slack head angle, which make it stable, helped us sail through rocks and roots and corner quickly. The tires can be run tubeless to better resist flats, and extra clearance means they spin freely when caked with mud. The CruX Elite is playful and fast — and something you won't outgrow as you collect medals.\n\nnspecialized.com $3,000\n\nACCESSORIES\n\nCraft Shield Glove\n\nThe cross season typically runs from September through February, so you'll need hearty gloves like these, with a fleece lining and a waterproof base, for warmth on wet race days. craftsports.us $78\n\nDarn Tough Micro Crew\n\nUnlike other bike races, cyclocross requires you to be on foot at times. So light, strong socks are key. These aren't likely to wear out, but Darn Tough will replace them if they do. darntough.com $18\n\nPark Tool Brush Set\n\nThe mud, dirt, and grime that builds up during off-road rides can damage key components. This kit does more than just keep your bike looking fresh; it keeps it healthy, too. parktool.com $80\n\nRapha Arm and Leg Warmer\n\nThese merino layers, which have a bit of Lycra for stretch, peel off easily when the weather warms up. And they dry quickly, whether you sweat profusely or get caught in a sudden squall. rapha.cc From $70\n\nTopeak SmartGauge D2\n\nFor peak performance, adjust your tire pressure to suit the terrain. (On soft trails, lower pressure makes it grip better.) The SmartGauge makes it a snap with a fast, readable result. topeak.com $40

Figure 6: **Demo case of Percpetion-R1** on OCR task.

