# OpenReview forum: "Perception-R1: Pioneering Perception Policy with Reinforcement Learning"
_NeurIPS.cc/2025/Conference — NeurIPS 2025 poster_

### Official Review · Reviewer_8gLg · 2025-06-06

**Clarity:** 2
**Significance:** 2
**Originality:** 3
**Rating:** 3
**Confidence:** 3

**Summary:**

Perception-R1 is a collections of Qwen VL 2 and 3B models finetuned using GRPO on a variety of vision tasks. The resulting models are performing well across a wide range of benchmarks, The authors provide interesting ablation on RL, SFT and SFT + RL and on the reward components.

**Questions:**

* You report the model achieving higher performance without thinking. How come the model cannot automatically learn to skip thinking if it is detrimental?

**Ethical Concerns:**

["NO or VERY MINOR ethics concerns only"]

**Final Justification:**

I am not in a position to recommend the paper in this current form. In my opinion, the main limitation of this work is that they trained a suite of specialize model, while the main challenge is to train a single model of a suite of tasks.

**Limitations:**

The author must acknowledge more explicitly that their method is limited to train specialist models since "the rewards for different tasks vary".

**Quality:**

2

**Strengths And Weaknesses:**

# Strengths

* If the multi subject reward function is novel, it is an interesting contribution.
* The degradation of the performance for Perception with thinking is interesting, but requires further investigation, i.e., why can't the policy learn to not think if it's detrimental to the performance?
* The authors test their method across a variety of tasks: grounding, counting, OCR, etc.

# Weaknesses

## Specialist model
> Notably, we do not mix the data from different perception tasks for joint training because the rewards for different tasks vary.
> We mainly use the Qwen2-VL-Instruct-2B as the baseline model.We also utilize Qwen2.5-VL-3B-Instruct [3] for training object detection tasks, due to its specialized optimization for localizing bounding boxes.
* It is not obvious from the text, but I understand that different models are trained for different tasks, instead of a single generalist model. This makes the results less impressive since the performance of Perception R1 is closer to the specialized baselines..

## Statistical significance and learning curves

* How was the RL checkpoint chosen for evaluation? The best performing checkpoint on the test set?
* Are the results consistent across seeds and training steps?

## Clarity:

* MLLM is used in the first sentence of the abstract and only defined on the 2nd page. Same for GRPO which is used at the end of the abstract.
* Please make it obvious that Perception-R1 is a suite of models and not a single generalist model.

## Missing prior work

* Some relevant important prior work is missing. For example,

Pinto, André Susano, et al. "Tuning computer vision models with task rewards." International Conference on Machine Learning. PMLR, 2023.

Mathe, Stefan, Aleksis Pirinen, and Cristian Sminchisescu. "Reinforcement learning for visual object detection." Proceedings of the IEEE conference on computer vision and pattern recognition. 2016.

---

> ### Author Rebuttal · Authors · 2025-07-31
>
> We sincerely appreciate the Reviewer for providing valuable feedback and suggestions. We have responded to each of your comments and questions as follows:
>
> * **Q1. Clarification and explanation regarding "specialist model”.**
>
> 	Thank you for your question regarding whether Perception-R1 is a single generalist model or a collection of specialist models. In fact, ***the core contribution of Perception-R1 is to propose a new, perception-first RL post-training paradigm.*** Our focus is more on the ***effectiveness*** and ***generality*** of this method rather than build a single unified model. For the sake of training efficiency, we trained different tasks separately using only a very small amount of data in our experiments. However, it is entirely feasible to mix the rewards from different tasks to perform joint training, and some recent work [1] has already done so. To further demonstrate this feasibility, we also conducted a simple mixed-reward experiment, as shown in the table below:
>
> 	| case | data | Visual Grounding | | | Visual Counting | |
>   |------|------|------------------|------------------|------------------| ------------------| ------------------|
>   | | |  RefCOCO | RefCOCO+ | RefCOCOg | Pixmo_val | Pixmo_test |
>   |single reward | 5k grounding | 89.1 | 81.7 | 85.7 | - | - |
>   |single reward| 5k counting | - | - | - | 78.1 | 75.6 |
>   |mix reward| 5k grounding + 5k counting | 88.6 | 81.1 | 85.2 | 78.0 | 75.2 |
>
> 	We can observe that mixing rewards from different tasks for joint training does not cause significant negative impact on task performance. Therefore, it is entirely feasible to train a single unified model if training efficiency is not a primary concern.
>
>
> 	[1] Ma, Yan, et al. "One RL to See Them All: Visual Triple Unified Reinforcement Learning."
>
> * **Q2. How was the RL checkpoint chosen for evaluation?**
>
> 	All experiments were evaluated using the final checkpoint of the trained model.
>
> * **Q3. Are the results consistent across seeds and training steps?**
>
> 	Regarding the training seed and the number of training steps, we compiled the model's performance under different training seeds and numbers of training steps, as shown in the table below:
>
> 	| exp | seed value | Visual Grounding | | |
>   |------|------|------------------|------------------|------------------|
>   | | |  RefCOCO | RefCOCO+ | RefCOCOg |
>   |1 | 123 | 88.7 | 81.0 | 85.1 |
>   | 2| 124 | 89.1 | 81.5 | 85.3 |
>   | 3| 125 | 88.8 | 81.0 | 85.4 |
>   | Average | -- | 88.9 | 81.2 | 85.3 |
>
> 	| exp | training step | Visual Grounding | | |
>   |------|------|------------------|------------------|------------------|
>   | | |  RefCOCO | RefCOCO+ | RefCOCOg |
>   | 4| 100| 88.5| 80.9| 84.5|
>   | 5| 200| 88.7| 81.3| 85.1|
>   | 6| 300| 88.8| 81.5| 85.7|
>   | last|357 | 89.1| 81.7| 85.7|
>
> 	we can observe that model training is not sensitive to seed settings (performance fluctuation does not exceed one point). Additionally, RL training steps show a positive correlation with final performance, and as training steps increase, performance gradually converges to an upper limit.
>
> * **Q4. Clarify of some abbreviations and missing prior work.**
>
> 	Thank you very much for your meticulous review. Based on your feedback, we have carefully examined all occurrences of technical term abbreviations to ensure that full explanations are provided upon first appearance. We have also added citations for the missing prior work you identified. These updates will be presented in the next version.
>
>
> * **Q5. Why not make model automatically learn to skip thinking?**
>
> 	Thank you for this interesting question. We would like to answer your query from two aspects:
>
> 	1. First, as our experiments in the paper demonstrate, an explicit linguistic thinking process is not beneficial for visual perception tasks. We observed that including such a process consistently degraded the model's final performance. This finding prompted us to disable the explicit linguistic thinking process and instead allow the model to learn an implicit "visual logic." For a more detailed explanation of "visual logic" and "linguistic logic," please refer to our response to **Q5** of Reviewer # **z4S9**.
>
> 	2. Second, it is indeed feasible to have the model autonomously learn to skip the thinking process. We conducted experiments on this as well (shown in the table below). Specifically, we enabled an explicit reasoning process training on the RefCOCO by incorporating `<think>` and `<answer>` special tokens. We then tracked the number of tokens generated within the `<think>` tags as training progressed. We can observe that during the RL process, the model tends to gradually shorten the length of its "thinking" tokens. However, this approach is highly inefficient, and the final performance is not as good as when we disable the thinking process directly.
>   | exp | training step | thinking_length |
>     |------|------|------------------|
>     | 1| 10 | 72|
>     | 2| 50 | 39|
>     | 3| 100| 27|
>     | 4| 150| 25|
>     | 5| 200| 20|
>
> 	Therefore, we opted to bypass the explicit reasoning steps, prompting the model to generate the final answer directly.

---

> > ### Author Response · Authors · 2025-08-05
> > **Kind Reminder to Reviewer 8gLg**
> >
> > Dear Reviewer 8gLg,
> >
> > We sincerely thank you for your valuable comments and suggestions on our work during this review process. We hope that our responses, including our individual reply to your feedback, and the responses to other reviewers, have addressed your concerns and questions. As the author rebuttal phase is coming to an end, we would like to kindly remind you to let us know if there are any remaining questions or issues you would like us to clarify further. We greatly value your insights and are more than happy to provide any further clarifications if needed to address any outstanding concerns.
> >
> > If your concerns have already been resolved, we would greatly appreciate your consideration of a higher rating, as this would play a significant role in the final evaluation of our work. Thank you again for your time and support!
> >
> > Best,
> > Authors

---

> > ### Comment · Reviewer_8gLg · 2025-08-05
> > **Response to Rebuttal by Authors**
> >
> > Thank you for your response.
> >
> > Re: Q1. Clarification and explanation regarding "specialist model”
> >
> > It is a bit disappointing that there are no gains obtained by training the model on different tasks.
> >
> > Re: Q5. Why not make model automatically learn to skip thinking?
> >
> > Interesting experiment. Thank you.

---

> > > ### Author Response · Authors · 2025-08-06
> > > **Follow-up the response to Reviewer 8gLg**
> > >
> > > We sincerely appreciate the reviewer’s further feedback and discussions. we are pleased our previous response addressed your concerns. Regarding your follow-up on **Q1** about why the mixed-reward approach did not yield additional gains over the single-reward setting, we offer two primary clarifications.
> > >
> > > * First, the experiment in **Q1** was a preliminary validation designed to confirm that our Perception-R1 framework could handle mixed-reward, multi-task training without significant degradation. The minor performance fluctuations observed (on the order of 0.x) were therefore within expectation. We view the current mixed-reward setting as a baseline with considerable room for optimization—more sophisticated reward balancing, shaping, and data mixture ratios could yield further gains.
> > > * Second, naively mixing tasks with heterogeneous properties, such as the inconsistent **label granularity** across grounding and counting, can introduce ambiguity. This makes it inherently challenging to surpass the performance of a model fine-tuned on a single, specific task and is a prevalent issue in MLLM training. We plan to mitigate this in future work through more sophisticated, granularity-aware data cleansing.
> > >
> > > Finally, we wish to re-emphasize that the "**surprising**" aspect of Perception-R1 lies in our novel approach of custom-designing the cognitive logic and reward systems for each perceptual task, which is essentially constructing a specific environment for each task. Our long-term vision is twofold. First, we aim to build a sufficiently large number of these perceptual environments to help the model "**learn to learn**." We believe that horizontally scaling the diversity of tasks will ultimately enhance the performance on each individual task (though this effect may start emerging at a certain point). Alternatively, we can scale vertically in depth, continuously modeling single-pass perceptual patterns through a "**try-error-retry**" paradigm powered by multi-turn interactions. We believe that this represents a vital direction for the next generation of multimodal reinforcement learning.

---

### Official Review · Reviewer_z4S9 · 2025-07-02

**Clarity:** 4
**Significance:** 3
**Originality:** 3
**Rating:** 5
**Confidence:** 4

**Summary:**

This paper introduces Perception-R1, a framework for enhancing the visual perception capabilities of Multimodal Large Language Models (MLLMs) through rule-based Reinforcement Learning (RL) during post-training. The paper argues that improving fundamental visual perception is a critical and currently under-explored prerequisite for achieving advanced visual reasoning. From the name, the paper positions itself as a visual-domain analogue to language-only reasoning models like DeepSeek-R1, by applying GRPO to Qwen2-VL series MLLM. The reward function uses task-specific, rule-based rewards such as IoU for visual grounding/object detection, and edit-distance for OCR tasks. For tasks with multiple objects, they used the Hungarian algorithm to solve the bipartite matching problem between prediction and ground truths.

The experiments showed 3 key findings: (1) an explicit CoT is found to be unnecessary and often detrimental to performance on the visual perception tasks studied (even though this was known to be helpful in language domain); (2) the effectiveness of RL over SFT is strongly correlated with the task's "perceptual perplexity," a measure of the output space's combinatorial complexity, with RL showing greater benefits on high-perplexity tasks like object detection; and (3) meticulous reward engineering, including multi-subject matching and a curriculum sampling strategy, is crucial for guiding the model toward learning robust perception policies.

In terms of empirical results, Perception-R1 achieved SOTA results on several perception benchmarks, including a +4.2% improvement on RefCOCO+, a +17.9% improvement on PixMo-Count, and a +4.2% F1-score on PageOCR. The most notable result is achieving 31.9% Average Precision (AP) on the COCO 2017 validation set, which the authors claim is the first time a pure MLLM architecture has reached this level of performance on this challenging object detection benchmark.

**Questions:**

1. On Formalism and Definitions: Could you provide a more formal RL-centric definition of a "perception policy"? Specifically, how do you define the state, action, and observation spaces for the MLLM agent in tasks like object detection?
2. On "Perceptual Perplexity": As mentioned above, how would you define perceptual perplexity with a formal equation?
3. On the Role of Chain-of-Thought: Your work finds that an explicit thinking process is detrimental for perception, which directly contrasts with the findings from DeepSeek-R1 that inspired your work. How do you explain this fundamental divergence? Does this suggest that visual perception and linguistic reasoning are governed by different cognitive mechanisms within these models that require distinct alignment strategies?
4. On the Choice of RL Algorithm: The paper uses GRPO, a policy-gradient algorithm. Could you justify this choice over preference-based optimization methods like DPO, or more specifically, PerPO, which targets the same problem of perceptual alignment in MLLMs? What are the hypothesized advantages of GRPO in this specific "perception policy" learning context?
5. On Attributing Performance Gains in COCO: The 31.9% AP on COCO is a very strong result. To help the community better understand the source of this gain, could you provide an ablation where Perception-R1 is applied to the non-specialized Qwen2-VL-2B model for the detection task? This would help disentangle the contribution of your RL framework from the strong localization priors of the Qwen2.5-VL base model.
6. On Statistical Significance: The paper reports impressive point-estimate improvements across all benchmarks. However, the checklist indicates that no error bars were reported. Could you please comment on the variance of your results across different training runs (e.g., with different random seeds) to establish the statistical significance of the claimed improvements?

**Ethical Concerns:**

["NO or VERY MINOR ethics concerns only"]

**Final Justification:**

I am increasing my rating for this paper. The authors' thorough rebuttal and the additional experiments they conducted have successfully addressed most of the significant concerns raised in my initial review.

Here is a summary of my final assessment:

* **Resolved: The critical issue of performance attribution.** My primary reservation was whether the impressive COCO result was due to the framework or the powerful base model. The new ablation applying Perception-R1 to the weaker Qwen2-VL-2B model convincingly demonstrated the framework's effectiveness, resolving this concern. This was the most important factor in my decision to raise the score.
* **Resolved: Lack of conceptual and empirical rigor.** The authors provided formal definitions for "perception policy" and "perceptual perplexity," which solidified the paper's conceptual foundations. The new experiments testing for statistical significance and generalization across tasks have also strengthened the paper's empirical claims.
* **Partially Resolved: Comparison to SOTA methods.** The authors provided a reasonable theoretical justification for their choice of GRPO over preference-based methods like DPO/PerPO. However, a direct empirical comparison is still missing. I view this as a minor remaining weakness, but not a critical flaw, as the primary comparison against SFT is sound and the paper's contribution is clear without it.

In summary, the authors' diligent response to feedback has significantly improved the paper. The paper now presents a well-supported and valuable contribution, with its core claims backed by much stronger evidence.

**Limitations:**

The authors provide a dedicated limitations section (Section 6), acknowledging that current perception tasks may be too simplistic to fully leverage RL's potential and that finding more complex "meta tasks" is a key future direction. However, the paper does not discuss the potential negative societal impacts of creating more powerful perception models. While the work is foundational, enhanced capabilities in object detection, OCR, and visual grounding could be applied to surveillance or autonomous systems with societal consequences. A brief discussion of these dual-use possibilities would strengthen this section.

**Quality:**

3

**Strengths And Weaknesses:**

**Strengths**:

**Originality and Significance**: On the surface it seems like a straightforward application of GRPO to RLVR CV tasks. Most attempts at applying RL to MLLMs followed a language-centric paradigm. This paper instead proposes a “perception-first” approach, with the core argument that robust perception is a prerequisite for high-level visual reasoning, is a valuable hypothesis. This reframes the problem from “how do we make MLLMs reason better?” to “how do we first make MLLMs see better, so that they can then reason?”.

**Quality**: The paper’s claims are supported by empirical validations across a diverse set of perception tasks. The result of 31.9% AP on COCO 2017 validation set is strong for MLLMs, although still behind expert models like Faster R-CNN (from 10 years ago), it is almost double that of the SFT baseline (16.1% AP). They also report consistent performance gains across visual grounding (RefCOCO+), visual counting (PixMo-Count), and dense OCR (PageOCR), which adds to the paper’s claim about enhancing foundational perception.

**Clarity and Reproducibility**: The paper is well-written and clearly organized. The experimental results includes sensible ablations, showing for example a clear performance degradation when removing reward matching (from 31.9 to 23.5 AP on COCO), supporting the need for bipartite matching mechanism. The counter-intuitive finding that explicit CoT harms performance is a well-supported. Finally, there is a detailed analysis of reward components.

**Weaknesses**:

1. **Lack of Conceptual Rigor**: Several key concepts central to the paper's thesis are not defined with sufficient precision, which limits their generalizability and testability.
  * The term "**Perception Policy**" is used throughout but its definition in Section 3 is a high-level, descriptive paragraph rather than a formal RL definition. The state space, action space (e.g., coordinate tokens, control tokens), and observation space for the MLLM agent are not formally specified. Without this formalism, the term remains ambiguous, making it difficult to precisely situate the work within the broader RL literature.
  * The concept of "**Perceptual Perplexity**" is introduced as a key determinant of RL's effectiveness. Having some sort of formal equation for perceptual perplexity and how Table 9 is computed would be more informative.
2. **Insufficient SOTA baselines**: The paper's positioning is narrow, primarily comparing its RL-based approach against SFT. It critically overlooks the most relevant and concurrent work on MLLM alignment, particularly methods based on preference optimization. Why is a policy-gradient approach like GRPO superior to, or different from, a preference-based approach for this specific problem?
  * One example that can be good to include for comparison is PerPO ([74] in the submission), which uses "discriminative rewarding" and "listwise preference optimization" to enhance the same visual discrimination capabilities in MLLMs and has results in RefCOCO/+/g as well (Table 1 in the submission).
3. **Limited Scope of Generalization Claims**: The paper trains task-specific models but makes claims about improving general capabilities. This claim is supported by limited evidence. The observation that RL training on the counting task improves performance on general comprehension benchmarks is intriguing but anecdotal. It is unclear if a model trained on object detection would also improve on ScienceQA, or if an OCR-trained model would improve on MM-Vet. Without a more systematic study, the claim of cross-task enhancement remains speculative.
4. **Potential Confounding Factors in Performance Gains**: The paper's strongest result (COCO detection) is achieved using a different, more powerful base model (Qwen2.5-VL-3B) than the other experiments. The Qwen2.5-VL model is explicitly designed and optimized for "precise object localization". While the ablation shows a large gain over the SFT version of this same model, it is difficult to disentangle the contribution of the Perception-R1 framework from the strong inductive biases of the base model's architecture. The claim would be more robust if a similar relative gain were demonstrated on a less-specialized base model, such as the Qwen2-VL-2B used for other tasks.

---

> ### Author Rebuttal · Authors · 2025-07-31
>
> We sincerely appreciate your recognition of our work, as well as your thoughtful and professional review. The suggestions you provided are highly valuable for the further improvement of our work. We have responded to each of your comments and questions as follows.
>
> * **Q1. Lack of conceptual rigor including Perception Policy and Perceptual Perplexity.**
>
> 	We are sorry that we did not provide detailed formal definitions in the main text due to the page limit. We offer the following clarifications below and will add them to the next revision.
>
> 	First, we aim to provide an RL-centric definition of **Perception Policy**. In the context of Perception-R1, the learned policy $\pi_{\theta}$ is the MLLM itself (e.g., Qwen2-VL). The learning process uses the GRPO algorithm to update the model’s parameters $\theta$ based on rule-based rewards calculated from generated outputs.
>
> 	During this process, the observation space $O$ can be defined as $O=(I,P)$, where $I$ is the image input, $P$ is the text-based prompt that specifies the task. For object detection, for example, the prompt is a request to output the bounding box coordinates and the class names of objects. The state at step $t$ can be formally defined as: $s_t = (O, a_1, a_2, ..., a_{t-1})$, where $(a_1, a_2, ..., a_{t-1})$ is the action sequence of tokens based on next-token prediction mechanism. For the specific task of object detection, the action sequence must conform to a strict structure encodes both object locations and their categories that looks like:`{"bbox_2d": [127, 54, 471, 406], "label": "laptop"}`. For the formal equation of **Perceptual Perplexity**, given a perceptual task dataset, the perceptual perplexity *P* of this task can be represented as:
>     $$P = \prod_{i=1}^{N} m_i!$$
> 	where $m_i$ is the number of ground-truth objects belonging to category $i$, $i$ ∈ {1, 2, ..., *N*} *N* is the total number of possible object categories for the task. The sum of objects across all categories must equal the total number of objects: $\sum_{i=1}^{N} m_i = M$. For visual grounding and OCR task, $M = N = 1$. For visual counting task, $M > 1, N = 1$. For object detection, $M > 1, N > 1$.
> * **Q2. Comparison between policy-gradient approach like GRPO and preference-based approach like DPO.**
>
> 	Thank you for your valuable suggestion. While the relative merits of policy-gradient RL like GRPO versus preference optimization like DPO have already been extensively studied in prior work [1][2][3], our work focuses primarily on their application within visual perception contexts. In this domain, we identify two principal advantages of GRPO over DPO:
> 	1. **Dynamic on-Policy exploration**. Through a dynamic and interactive training process, GRPO samples data from the current policy model and continuously updates the policy online. This enables it to explore the policy space, learn from its own mistakes, adapt to its evolving capabilities, and constantly create new, more challenging "curricula" for itself. This implies that GRPO affords the model a broader exploration space, enabling it to discover superior visual perception polices. In contrast, DPO, as an offline method, relies on a fixed preference dataset, which leads to a "distribution shift" problem and limits its further performance improvement.
> 	2. **Generation of high-quality hard negatives.** GRPO's online sampling naturally generates high-quality negative examples that are relevant and challenging. Since the model explores at the edge of its capabilities, its failures are meaningful mistakes rather than random errors. These "hard negatives" are incorrect but plausible solutions that provide maximum learning value. DPO is limited by its pre-collected dataset, where the quality of negative examples may not target the model's specific weaknesses as it evolves. GRPO therefore creates a more effective learning process by continuously finding its own most challenging examples.
>
> 	As for the work on PerPO [4], although it is specifically optimized for visual perception tasks through "discriminative rewarding" and "listwise preference optimization”, its approach still falls within the paradigm of DPO. Consequently, it cannot fundamentally resolve the two aforementioned deficiencies compared to GRPO.
>
> 	[1] Wang, Bo, et al. "Implicit Reward as the Bridge: A Unified View of SFT and DPO Connections."
> 	[2] Tang, Yunhao, et al. "Understanding the performance gap between online and offline alignment algorithms."
> 	[3] Lanchantin, Jack, et al. "Bridging Offline and Online Reinforcement Learning for LLMs."
> 	[4] Zhu, Zining, et al. "Perpo: Perceptual preference optimization via discriminative rewarding."
>
> * **Q3. More systematic study to support the claim of generalization enhancement.**
>
> 	Based on your suggestion, we ran additional experiments to test our generalization claim. We used models post-trained on different tasks. Note that we excluded OCR because its focus on text extraction differs fundamentally from the other tasks, which require general visual capabilities like spatial perception and object identification. The results are presented below.
>
> 	| | post-training task | MMBench | MMVet | MMStar | ScienceQA | SeedBench | MME | | LLaVA-Bench | AI2D |
>   |---|---|---|---|---|---|---|---|---|---|---|
>   | | | Avg | Avg | Avg | Avg | Avg | Cognition | Perception | Avg | Avg |
>   | Qwen2-VL-2B | -- | 71.9 | 45.6 | 46.3 | 74.0 | 72.7 | 418.5 | 1471.1 | 46.5 | 71.6 |
>   | **Perception-R1** | visual counting | 71.8 | 48.9 | 45.7 | 73.4 | 73.0 | 430.0 | 1473.9 | 58.2 | 71.8 |
>   | | visual grounding | 71.4 | 49.2 | 46.6 | 73.6 | 72.9 | 425.4 | 1457.2 | 52.0 | 71.3 |
>   | Qwen2.5-VL-3B | -- | 71.4 | 57.7 | 53.3 | 75.8 | 70.9 | 560.7 | 1470.8 | 62.7 | 79.0 |
>   | | object detection | 78.5 | 56.4 | 55.7 | 78.1 | 73.7 | 611.4 | 1546.7 | 65.2 | 80.5 |
>
> 	The results show that the Perception-R1, when trained on diverse visual tasks, consistently enhances general visual understanding and reasoning. This demonstrates the effectiveness and generality of our method.
>
> * **Q4. More experimental results based on Qwen2-VL and Qwen2.5-VL.**
>
> 	In response to your suggestion, we have conducted additional experiments comparing our Perception-R1 framework with the Qwen2-VL and Qwen2.5-VL baselines. The experimental results are presented as follows:
>
> 	| case | base model | Visual Grounding | | | OCR | Visual Counting | | Detection |
>   |------|------|------------------|------------------|------------------|---------|------------------|------------------|-----------|
>   | | |  RefCOCO | RefCOCO+ | RefCOCOg | PageOCR | Pixmo_val | Pixmo_test | COCO2017 |
>   | Perception -R1 | Qwen2-VL-2B | 89.1 | 81.7 | 85.7 | **98.2** | 78.1 | 75.6 | 28.0|
>   | | Qwen2.5-VL-3B | **89.5** | **83.5** | **86.5** |97.1 | **83.8**| **81.1**| **31.9**|
>
>
> * **Q5. Does the difference in reasoning between Perception-R1 and DeepSeek-R1 imply that their visual and linguistic abilities rely on separate mechanisms, each requiring its own alignment strategy?**
>
> 	Yes, as we argue in the paper, this difference arises because visual perception tasks are primarily oriented toward visual logic rather than semantic or linguistic logic. For tasks like object counting or localization, a model learns implicit patterns directly from image pixel. Imposing an explicit, language-based reasoning chain introduces an unnecessary and potentially detrimental step that degrades visual perception capabilities. Similar findings are also documented in [1].
>
> 	Furthermore, we wish to clarify that although the reasoning paradigms for visual perception and linguistic reasoning differ, they can still be unified. By designing and combining distinct reward functions, a model can be trained to accommodate both modes of thinking, an avenue that has already been explored in [2].
>
> 	[1] Wei, Yana, et al. "Open Vision Reasoner: Transferring Linguistic Cognitive Behavior for Visual Reasoning."
> 	[2] Ma, Yan, et al. "One RL to See Them All: Visual Triple Unified Reinforcement Learning."
>
> * **Q6. Provide statistical significance such as with different random seeds.**
>
> 	We appreciate you highlighting the importance of statistical significance. In response, we conducted stability experiments for Perception-R1 on the RefCOCO/+/g. Each experiment utilized a different random seed to assess the robustness of our findings, and the results are shown below.
>
> 	| case | seed value | Visual Grounding | | |
>   |------|------|------------------|------------------|------------------|
>   | | |  RefCOCO | RefCOCO+ | RefCOCOg |
>   |1 | 123 | 88.7 | 81.0 | 85.1 |
>   | 2| 124 | 89.1 | 81.5 | 85.3 |
>   | 3| 125 | 88.8 | 81.0 | 85.4 |
>   | Average | -- | 88.9 | 81.2 | 85.3 |
>
> 	We can observe that under different random seed settings, the model's performance fluctuation on benchmarks does not exceed 1 point, indicating that Perception-R1 possesses sufficient stability to support reproducibility.
> * **Q7. Negative societal impacts of creating more powerful perception models.**
>
> 	We appreciate you raising the critical issue of societal impact. We agree this was an important omission and, following your suggestion, have added a discussion about this as follows:
> 	“It is essential to acknowledge the broader societal impacts and dual-use nature of our work. Like many AI technologies, the perception models that offer benefits in visual grounding, OCR, and object detection can also be exploited. For instance, highly accurate person detection could fuel mass surveillance and threaten civil liberties, while powerful OCR could scan sensitive documents without consent. As these models become foundational to AI systems, ethical considerations become paramount. We urge the community to lead the conversation on responsible development, establish clear guidelines, and create robust safeguards to mitigate these risks.”

---

> > ### Comment · Reviewer_z4S9 · 2025-08-06
> >
> > Thank you for the detailed rebuttal and for running the additional experiments to address the reviewers' questions.
> > Your responses have helped to clarify several points from my initial review:
> > 1. The provided definitions for the "perception policy" and "perceptual perplexity" are helpful in formalizing the paper's core concepts.
> > 2. The new cross-task evaluation provides some initial evidence for the framework's generalization benefits, which is a useful addition.
> > 3. I appreciate you running the requested COCO detection ablation on the Qwen2-VL-2B model. The reported result is a substantial improvement over the baseline and helps address the concern about disentangling the framework's contribution from the base model's intrinsic capabilities.
> > 4. Your justification for using GRPO over preference-based methods like DPO is reasonable, though an empirical comparison remains an interesting direction for future work.
> >
> > I will take these clarifications and the new data into account in my final assessment of the paper.

---

> > > ### Author Response · Authors · 2025-08-06
> > > **Gratitude for the positive feedback of Reviewer z4S9**
> > >
> > > Dear Reviewer z4S9,
> > >
> > > We sincerely appreciate your further feedback and your positive reception of our rebuttal! Thank you for your valuable comments and suggestions on our work. All the clarifications and additional experiments mentioned in the rebuttal will be incorporated into the revision of our manuscript.
> > >
> > > Finally, we would like to thank you once again for your thorough and professional review of our work.
> > >
> > > Best,
> > > Authors

---

### Official Review · Reviewer_AEua · 2025-07-03

**Clarity:** 3
**Significance:** 3
**Originality:** 3
**Rating:** 4
**Confidence:** 4

**Summary:**

This paper proposes Perception-R1, a rule-based reinforcement learning (RL) framework aimed at improving visual perception capabilities in multimodal large language models (MLLMs). Inspired by the success of DeepSeek-R1 in pure language domains, the authors apply the GRPO algorithm during the post-training phase of MLLMs to optimize perception-oriented tasks like visual grounding, counting, OCR, and object detection.

Key findings include: 1) Explicit "thinking" processes (e.g., Chain-of-Thought) hurt perception tasks. 2) Reward design is crucial, especially in multi-object tasks. 3) RL benefits tasks with high perceptual perplexity (e.g., object detection) but less so in simpler tasks (e.g., OCR). 4) Perception-R1 achieves strong or state-of-the-art results across multiple benchmarks, especially hitting 31.9 mAP on COCO2017 — the first for a pure MLLM.

**Questions:**

*  Most benchmarks are academic; demonstrating robustness in noisy or real-world environments would enhance impact. Therefore, limited evaluation on real-world applications is a big concern for me.
* The paper doesn’t state whether code, models, or data subsets will be released, which is key for community adoption.
* **Why SFT + RL Performs Worse than RL?** SFT + RL is often the common practice of many previous research efforts.

I appreciate the overall pipeline and the proposed method for incorporating reinforcement learning into training grounded multimodal LLMs. However, I have some concerns about the potentially overreaching title and naming choices (e.g., Perception-R1), which may give the impression of a more comprehensive coverage of perception tasks than what is actually explored in the paper. Also, I think it is necessary to perform experiments on more than one base models to demonstrate the generalizability of the proposed method.

**Ethical Concerns:**

["NO or VERY MINOR ethics concerns only"]

**Final Justification:**

I appreciate the authors addressing my questions in the rebuttal. I like the core idea of extending the findings in DeepSeek-R1 from the purely language domain to visual perception. The paper applies GRPO in the post-training stage of MLLMs to optimize perception-oriented tasks such as visual grounding, counting, OCR, and object detection. Therefore, I will maintain my positive score and recommend acceptance.

**Limitations:**

Yes

**Quality:**

3

**Strengths And Weaknesses:**

### **Strengths**

* Offers a new angle by targeting perception-first reinforcement learning, rather than language-first reasoning, for MLLMs.
* Implements GRPO without complex scaffolding (e.g., MCTS), making the approach scalable and accessible.
* State-of-the-art results: Sets new benchmarks on visual grounding (RefCOCO/+), counting (PixMo), OCR (PageOCR), and general object detection (COCO2017).
* Detailed Ablations & analysis: Provides thorough studies on reward design, the (in)effectiveness of thinking processes, scaling rollouts, and task complexity.

### **Weaknesses / Questions**

* The name Perception-R1 suggests a general-purpose perception policy framework, but the main performance gains and experimental focus appear to be centered on object detection. Could the authors clarify whether the method generalizes across a broader range of perception tasks (e.g., segmentation, depth estimation, tracking), or is it primarily tailored for detection and counting?
* While results show GRPO helps, there's little theoretical grounding as to *why* rule-based RL is well-suited for visual tasks beyond empirical trends.
* Most experiments are built on Qwen2-VL; broader model diversity (e.g., LLaVA, MiniGPT) might improve generality of claims. Overreliance on one base model.
* While the reward complexity claimed to be simple, reward functions are still tailored per task and depend on carefully tuned heuristics, which may not scale easily.

---

> ### Author Rebuttal · Authors · 2025-07-31
>
> We greatly appreciate the Reviewer for the recognition of our work and the valuable comments and suggestions you provided. We have responded to each of your comments and questions as follows.
>
> * **Q1. The "Perception-R1" name and its scope for broader perception tasks.**
>
> 	Thanks for your suggestion. We would like to emphasize that ***the core contribution of Perception-R1 is to propose a new, perception-first RL post-training paradigm.*** So we name our framework “Perception-R1”. To demonstrate its effectiveness, we selected four foundational visual perception tasks: visual grounding, OCR, visual counting, and object detection. Theoretically, our framework can be adapted to any visual perception task by defining an appropriate reward function. In fact, some latest works have already attempted to use rule-based RL techniques for segmentation [1] and tracking [2], which can be viewed as extensions of our work. In the future, we will also explore applying Perception-R1 to 3D visual perception tasks (such as 3D object detection and tracking, depth estimation, etc.) to further expand its application scope.
>
> 	[1] You, Zuyao, and Zuxuan Wu. "Seg-R1: Segmentation Can Be Surprisingly Simple with Reinforcement Learning."
> 	[2] Wang, Biao, and Wenwen Li. "R1-track: Direct application of mllms to visual object tracking via reinforcement learning."
>
> * **Q2. Theoretical explanation of why rule-based RL is well-suited for visual tasks.**
>
> 	This question is similar to the **Q1** of Reviewer # **S8da**. To further elaborate on our reasoning, we highlight the following points:
> 	 1. **RL conducts better negative supervision compared to SFT.** Visual representation learning and downstream visual task like *object detection* are highly dependent on the quantity and quality of *negative samples* [1][2]. In the context of post-training, the negative sampling spaces and strategies for RL and SFT differ significantly. With RL, any action that yields a low reward is treated as a "negative sample." This allows RL model to effectively explore and learn from a wide range of trajectory samples on-line. While SFT methods only sample one trajectory (answer text) off-line and the negative supervision is very limited (only from a few tokens that are not strictly aligned with the answer).
> 	 2. **Policy gradient allows models to explore the perception pattern freely not just clone the expert behavior.** SFT is fundamentally a form of behavior cloning (BC), where the model learns to maximize the likelihood of an expert's demonstration (the ground-truth text). This approach can suffer from covariate shift: when the model makes a mistake, it enters a state distribution not seen during training, leading to compounding errors. In contrast, RL, particularly through policy gradient methods, optimizes the policy $\pi$ to maximize the expected cumulative reward. This objective encourages the model to explore and discover novel strategies or implicit ”perception patterns" that lead to high rewards, even if those patterns deviate from the expert's specific demonstration. It learns to solve the task robustly, rather than simply mimicking one particular solution path.
> 	 3. **RL allows for more flexible and task-aligned reward shaping.** The objective for SFT is fixed: minimizing the cross-entropy loss against a reference text. This is an indirect proxy for the actual task performance. For many visual tasks, performance is better measured by non-differentiable or complex metrics. RL's reward function $R$ can be designed to directly incorporate these metrics. For instance, in object detection, the reward can be the Intersection over Union (IoU) score, directly optimizing for localization accuracy. For image editing, the reward could be based on a combination of a CLIP score for semantic alignment and a score from an aesthetic quality model. This flexibility allows RL to align the model's learning process much more closely with the true end-goal of the visual task.
>
> 	[1] He, Kaiming, et al. "Momentum contrast for unsupervised visual representation learning."
> 	[2] Lin, Tsung-Yi, et al. "Focal loss for dense object detection."
>
> * **Q3. More baseline results.**
>
> 	Following your advice, we add more baselines including Qwen2-VL, Qwen2.5-VL, and LLaVA to apply our Perception-R1 framework. The results are shown as following:
>
> 	| case | base model | Visual Grounding | | | OCR | Visual Counting | | Detection |
>   |------|------|------------------|------------------|------------------|---------|------------------|------------------|-----------|
>   | | |  RefCOCO | RefCOCO+ | RefCOCOg | PageOCR | Pixmo_val | Pixmo_test | COCO2017 |
>   | LLaVA-1.5-7B | - | 57.2 | 49.8 | 50.2 | - | - | - | - |
>   | Perception -R1 | LLaVA-1.5-7B | 61.8| 54.0| 55.6| - | - | - | -|
>   | Qwen2-VL-2B | - | 86.8 | 77.1 | 83.3 | 94.4 | 60.2 | 50.5 | 0.1 |
>   | Perception -R1 | Qwen2-VL-2B | 89.1 | 81.7 | 85.7 | **98.2** | 78.1 | 75.6 | 28.0|
>   | Qwen2.5-VL-3B | - | 86.7 | 79.3 | 84.3 | 94.9 | 60.8 | 57.4 | 16.1 |
>   | Perception-R1 | Qwen2.5-VL-3B | **89.5** | **83.5** | **86.5** | 97.1 | **83.8**| **81.1**| **31.9**|
>
>
> * **Q4. Reward functions are still tailored per task and depend on carefully tuned heuristics, which may not scale easily.**
>
> 	Thank you for your concerns regarding the reward function design. The reward function is central to rule-based RL. Given the discriminative nature of visual perception tasks, most reward functions are discriminative metrics such as Intersection over Union (IoU) or Euclidean distance. These different task-specific reward functions are not mutually exclusive, rather, they can be mixed and scaled up. Recent work [1] has explored this approach, demonstrating the effectiveness and scalability of a unified RL framework for multiple downstream tasks.
>
> 	Furthermore, from an RL perspective, this multi-component reward structure functions can be regarded as task-aligned reward shaping. As discussed in the response of **Q2**, for many visual tasks, performance is better measured by non-differentiable or complex metrics. RL's reward function $R$ can be designed to directly incorporate these metrics. By providing a dense and multifaceted objective rather than a single sparse signal, it inherently mitigates the risk of policy exploitation (i.e., reward hacking), thereby compelling the model to learn more robust and generalizable policies.
>
> 	[1] Ma, Yan, et al. "One RL to See Them All: Visual Triple Unified Reinforcement Learning."
>
>
> * **Q5. Limited evaluation on real-world applications.**
>
> 	Thanks for your advice. To address your concerns, we have conducted additional evaluations in real-world and robotic scenarios. Specifically, we select two challenging real-world understanding and reasoning benchmarks, i.e., RealWorldQA, and OmniSpatial [1] to evaluate Perception-R1.  The results are presented in the table below.
>
> 	| models | RealWorldQA | OmniSpatial |
>   |--------|-------------|-------------|
>   | Baseline (Qwen2.5-VL-3B) | 60.26 | 41.81 |
>   | Perception-R1 | **61.83** | **42.07** |
>
> 	We can observe that Perception-R1 still demonstrates its effectiveness in noisy real-world scenarios.
>
>
> 	[1] Jia, Mengdi, et al. "OmniSpatial: Towards Comprehensive Spatial Reasoning Benchmark for Vision Language Models."
>
> * **Q6. Open code and model weights.**
>
> 	Thank you for this valuable suggestion. In fact, we are already prepared to make our work open-source including data, code, and model weight. In order to adhere to the NeurIPS double-blind review policy, we have omitted this information in the current submission. We will be sure to include a direct link to the open-source repository in the final camera-ready version.
>
> * **Q7. Why SFT + RL performs worse than RL-only?**
>
> 	Thank you for your question. We attribute the reason to two primary factors:
>
> 	1. Our base model, i.e., Qwen2-VL and Qwen2.5-VL, have already undergone a comprehensive SFT stage. Applying an additional SFT phase on this foundation risks disrupting the model's originally learned data distribution.
> 	2. As we analyze in the response for **Q2**, SFT is fundamentally a form of behavior cloning (BC) [1]. This suggests the model can become constrained by the rigid mimetic patterns acquired during the SFT phase, thereby impeding effective exploration in the subsequent RL stage.
>
> 	As a result, performance can actually degrade, making the direct application of RL a more effective approach.
>
> 	[1] Chen, Jierun, et al. "The Synergy Dilemma of Long-CoT SFT and RL: Investigating Post-Training Techniques for Reasoning VLMs.”

---

> > ### Comment · Reviewer_AEua · 2025-08-06
> >
> > Thank you to the authors for addressing most of my concerns in the rebuttal. I also appreciate the additional experiments conducted on more base models.
> >
> > I have two follow-up questions regarding your responses to Q1 and Q7:
> >
> > Q1: While I appreciate your clarification that some papers follow the Perception-R1 approach for segmentation and tracking, these tasks are still fundamentally localization-type tasks. This does not provide sufficient evidence that the proposed method can be effectively applied to tasks beyond detection/segmentation. Because of this, I feel that Perception-R1 may not be the most appropriate name, as it suggests a broader scope than what is actually covered in this paper.
> >
> > Additionally, the reward model design in this work is heavily tailored to localization. In RL, reward model design is often one of the most critical components (perhaps even the most critical), which deepens my concern about whether the naming accurately reflects the method’s broader applicability.
> >
> > Q7: You mentioned that "applying an additional SFT phase on this foundation risks disrupting the model's originally learned data distribution." However, wouldn’t training the model with RL also disrupt the original learned data distribution? Could you clarify this point further?
> >
> > Overall, as I mentioned in my initial review, I find the core idea of introducing RL for perception tasks in generative models to be compelling. Therefore, I will maintain my initial positive score. But I do have strong concerns on generalizability of the reward models used in this paper, which is hard to be applied to tasks unlike localization tasks.

---

> > > ### Author Response · Authors · 2025-08-07
> > > **Follow-up response to Reviewer AEua**
> > >
> > > We are very grateful for the reviewer's positive feedback on our rebuttal and for the opportunity for further discussion. In response to the two concerns raised regarding our replies to **Q1** and **Q7**, we would like to offer the following further clarifications and explanations.
> > >
> > > * **Q1. Further discussion on works related to Perception-R1 and the scope of its naming.**
> > >
> > > 	We appreciate the reviewer's further discussion on the scope and applicability of Perception-R1. To clarify our response to **Q1**, we cited works on segmentation and tracking primarily because the reviewer's question explicitly mentioned these two tasks. We wish to emphasize that the ***Perception-R1 paradigm is not exclusively applicable to localization tasks***. In fact, it can also address perception tasks unrelated to localization, such as *object classificatio*n [1] and *OCR* (Table 2 in the manuscript). Furthermore, latest work [2] has already applied this paradigm to *3D perception*. Therefore, the Perception-R1 paradigm can be adapted to nearly all visual perception tasks with clear reward, and in this paper, we selected four of the most representative foundational tasks to demonstrate this versatility.
> > >
> > > 	Furthermore, we strongly agree with the reviewer on the importance of the reward design. In fact, the reward modeling presented in this paper serves as a proof-of-concept to illustrate our core philosophy. The core insight of Perception-R1 lies in our novel approach of custom-designing the cognitive logic and reward systems for each perceptual task, which is essentially constructing a specific environment for each task. Our long-term vision is twofold. First, we aim to build a sufficiently large number of these perceptual environments to help the model '**learn to learn**.' We believe that horizontally scaling the diversity of tasks will ultimately enhance the performance on each individual task (though this effect may start emerging at a certain point). Alternatively, we can scale vertically in depth, continuously modeling single-pass perceptual patterns through a '**try-error-retry**' paradigm powered by multi-turn interactions. We believe that this represents a vital direction for the next generation of multimodal reinforcement learning.
> > >
> > > 	[1] Li, Ming, et al. "Think or not think: A study of explicit thinking in rule-based visual reinforcement fine-tuning."
> > > 	[2] Huang, Ting, Zeyu Zhang, and Hao Tang. "3D-R1: Enhancing Reasoning in 3D VLMs for Unified Scene Understanding."
> > >
> > >
> > > * **Q7. Further explanation on why RL is less disruptive to the base model's learned distribution compared to SFT.**
> > >
> > > 	Thank you for your further inquiry on this question. We are pleased to provide additional clarification. In fact, a significant body of research [1] has already investigated the impact of SFT and RL on the learned distribution of base models. Among the conclusions drawn, the most critical reason is:
> > >
> > > 	Modern RL fine-tuning, particularly through established algorithms like Proximal Policy Optimization (PPO) [2] and GRPO, incorporates a ***KL divergence penalty*** into its objective function. This term mathematically measures and constrains how much the policy's output distribution can deviate from the original model's distribution. This constraint acts as a regularization "leash," ensuring the model can learn new aligned behaviors while mitigating the risk of catastrophically forgetting its vast, pre-existing knowledge. SFT protocols typically lack this explicit safeguard, making them more susceptible to significant and potentially detrimental distributional shifts.
> > >
> > > 	[1] Li, Hongyu, et al. "Revisiting catastrophic forgetting in large language model tuning."
> > > 	[2] Schulman, John, et al. "Proximal policy optimization algorithms."

---

### Official Review · Reviewer_S8da · 2025-07-06

**Clarity:** 3
**Significance:** 3
**Originality:** 3
**Rating:** 4
**Confidence:** 4

**Summary:**

Perception-R1 introduces a RLVR framework that fine-tunes multimodal LLMs for perception tasks including visual grounding, OCR, counting and object detection. The authors argue that when paired with carefully designed rewards (including their reward-matching scheme) and no chain-of-thought reasoning, RL can unlock stronger visual skills for multimodal LLMs. A key element is a reward-matching step that aligns each predicted object with the most appropriate ground-truth instance before scoring, ensuring the model is rewarded for true perceptual accuracy rather than output order. Using only 5k–10k samples per task, this approach shows superior performance compared to baselines.

**Questions:**

- Why—and under what inductive bias—can RL outperform SFT when no chain-of-thought reasoning is used? In other words, if explicit reasoning is absent, what advantage does a policy-gradient objective confer that standard maximum-likelihood training cannot. Does this also imply that training an encoder with RL instead of SL would also obtain superior simple image-classification tasks?
- In Tables 1–4, are the MLLM scores zero-shot or supervised fine-tuned results? If any numbers are zero-shot, please also report supervised-only results so readers can fairly assess the benefit of RL tuning. Also please including the Qwen-2.5-VL backbone results for fair comparison. Perception-R1 uses Qwen2.5-VL selectively.
- Table 5 shows the “no RL” baseline outperforming “SFT only.” What training differences separate these rows, and if "no RL" baselines indicates a zero-shot baseline (a baseline with no training) why would additional supervised training lower accuracy for certain tasks?
- At line 256-257, the authors state that "for low-perplexity tasks like grounding and OCR, RL performs comparably to or even worse than SFT", but I don't see any values where RL outperforms SFT (row 6 vs row 7). What is the reason behind this?

**Ethical Concerns:**

["NO or VERY MINOR ethics concerns only"]

**Final Justification:**

The reviewers has resolved most of my concerns and hence, raise my score to Borderline Accept.

**Limitations:**

yes

**Quality:**

3

**Strengths And Weaknesses:**

Strengths
- The paper is one of the first to study RL post-training for core visual perception in MLLMs, filling a gap left by language-centric RL work.
- The authors propose a simple yet effective GRPO based recipe which shows consistent gains across four perception benchmarks.
- The study provide ablation studies which isolates three important lessons (no CoT, value of reward matching, role of perceptual perplexity) that can guide future RL practitioner later in this field.
- The paper is overall easy to follow.

Weaknesses
- Beyond empirical wins on high-perplexity tasks, the paper offers little theoretical reason RL should outperform SFT once the thinking component is removed from the post-training stage.
- The definition and justification of the perceptual-perplexity metric are opaque.
- Several comparison and reporting issues flagged in the questions section need clarification for full transparency.

---

> ### Author Rebuttal · Authors · 2025-07-31
>
> Thank you very much for recognizing our work and for providing such valuable comments and suggestions. We have responded to each of your comments and questions as follows:
> * **Q1. Theoretical explanation about why RL outperforms SFT without CoT process.**
>
> 	Thanks for your question. This is a profound and valuable topic. Indeed, some existing works [1][2][3] have already explored the relative merits of RL and SFT. In the context of visual perception, we think the following key reasons are crucial:
> 	 1. **RL conducts better negative supervision compared to SFT.** Visual representation learning and downstream visual task like *object detection* are highly dependent on the quantity and quality of *negative samples* [4][5]. In the context of post-training, the negative sampling spaces and strategies for RL and SFT differ significantly. With RL, any action that yields a low reward is treated as a "negative sample." This allows RL model to effectively explore and learn from a wide range of trajectory samples on-line. While SFT methods only sample one trajectory (answer text) off-line and the negative supervision is very limited (only from a few tokens that are not strictly aligned with the answer).
> 	 2. **Policy gradient allows models to explore the perception pattern freely not just clone the expert behavior.** SFT is fundamentally a form of behavior cloning (BC), where the model learns to maximize the likelihood of an expert's demonstration (the ground-truth text). This approach can suffer from covariate shift: when the model makes a mistake, it enters a state distribution not seen during training, leading to compounding errors. In contrast, RL, particularly through policy gradient methods, optimizes the policy $\pi$ to maximize the expected cumulative reward. This objective encourages the model to explore and discover novel strategies or implicit ”perception patterns" that lead to high rewards, even if those patterns deviate from the expert's specific demonstration. It learns to solve the task robustly, rather than simply mimicking one particular solution path.
> 	 3. **RL allows for more flexible and task-aligned reward shaping.** The objective for SFT is fixed: minimizing the cross-entropy loss against a reference text. This is an indirect proxy for the actual task performance. For many visual tasks, performance is better measured by non-differentiable or complex metrics. RL's reward function $R$ can be designed to directly incorporate these metrics. For instance, in object detection, the reward can be the Intersection over Union (IoU) score, directly optimizing for localization accuracy. For image editing, the reward could be based on a combination of a CLIP score for semantic alignment and a score from an aesthetic quality model. This flexibility allows RL to align the model's learning process much more closely with the true end-goal of the visual task.
>
>
> 	And this conclusion still holds true for the simple image classification task [6].
>
>
> 	[1] Chen, Jierun, et al. "The Synergy Dilemma of Long-CoT SFT and RL: Investigating Post-Training Techniques for Reasoning VLMs.”
> 	[2] Zhu, Ke, et al. "Continual sft matches multimodal rlhf with negative supervision."
> 	[3] Wang, Bo, et al. "Implicit Reward as the Bridge: A Unified View of SFT and DPO Connections."
> 	[4] He, Kaiming, et al. "Momentum contrast for unsupervised visual representation learning."
> 	[5] Lin, Tsung-Yi, et al. "Focal loss for dense object detection."
> 	[6] Li, Ming, et al. "Think or not think: A study of explicit thinking in rule-based visual reinforcement fine-tuning."
>
> * **Q2. The definition and justification of the perceptual-perplexity metric.**
>
> 	We are sorry that we did not provide detailed formal definitions in the main text due to the page limit. We offer the following clarifications of perceptual perplexity below and will add them to the next revision.
> 	**Definition:** Given a perceptual task dataset, the perceptual perplexity *P* of this task can be represented as:
> 	$$P = \prod_{i=1}^{N} m_i!$$
> 	where $m_i$ is the number of ground-truth objects belonging to category $i$, $i$ ∈ {1, 2, ..., *N*} *N* is the total number of possible object categories for the task. The sum of objects across all categories must equal the total number of objects: $\sum_{i=1}^{N} m_i = M$.
> 	**Justification:** For visual grounding and OCR task, $M = N = 1$. For visual counting task, $M > 1, N = 1$. For object detection, $M > 1, N > 1$. Consequently, the perceptual perplexity associated with these tasks increases from low to high.
>
> * **Q3. Experiment setting in Table 1-4 and add Qwen2.5-VL baseline result.**
>
> 	The results for Perception-R1 presented in Tables 1-4 were obtained in a zero-shot setting. Concurrently, we report the SFT-only results in Table 5. Following your advice, we have added the results of Qwen2.5-VL as a baseline, as shown in the table below. These results will be included in the next version.
>
> 	| case | base model | Visual Grounding | | | OCR | Visual Counting | | Detection |
>   |------|------|------------------|------------------|------------------|---------|------------------|------------------|-----------|
>   | | |  RefCOCO | RefCOCO+ | RefCOCOg | PageOCR | Pixmo_val | Pixmo_test | COCO2017 |
>   | Qwen2-VL-2B | - | 86.8 | 77.1 | 83.3 | 94.4 | 60.2 | 50.5 | 0.1 |
>   | Qwen2.5-VL-3B | - | 86.7 | 79.3 | 84.3 | 94.9 | 60.8 | 57.4 | 16.1 |
>   | Perception -R1 | Qwen2-VL-2B | 89.1 | 81.7 | 85.7 | **98.2** | 78.1 | 75.6 | 28.0|
>   | Perception-R1 | Qwen2.5-VL-3B | **89.5** | **83.5** | **86.5** | 97.1 | **83.8**| **81.1**| **31.9** |
>
>
> * **Q4. Issue of the comparison between SFT-only and baseline setting in Table. 5.**
>
> 	We sincerely apologize that there is an oversight in the OCR column of Table 5. In fact, the correct baseline metric for row 3 in the OCR column is 94.4, which is consistent with the Qwen2-VL baseline reported in Table 3. With this correction, it is clear that the performance in the SFT-only setting is generally higher than the baseline performance. We will rectify this in the revised manuscript.
>
> * **Q5. Confusion about the statement at line 256-257.**
>
> 	Thank you for pointing out the ambiguity of this sentence. To clarify, our intention is to convey that for tasks with low perceptual perplexity, such as visual grounding and OCR, the RL-only setting does not yield a significant performance improvement over the SFT-only method. The gain is only about one point, which is why we describe the results as "comparable”. In the next version, we will revise this statement to be more precise and avoid any ambiguity.

---

> > ### Author Response · Authors · 2025-08-05
> > **Kind Reminder to Reviewer S8da**
> >
> > Dear Reviewer S8da,
> >
> > We sincerely thank you for your valuable comments and suggestions on our work during this review process. We hope that our responses, including our individual reply to your feedback, and the responses to other reviewers, have addressed your concerns and questions. As the author rebuttal phase is coming to an end, we would like to kindly remind you to let us know if there are any remaining questions or issues you would like us to clarify further. We greatly value your insights and are more than happy to provide any further clarifications if needed to address any outstanding concerns.
> >
> > If your concerns have already been resolved, we would greatly appreciate your consideration of a higher rating, as this would play a significant role in the final evaluation of our work. Thank you again for your time and support!
> >
> > Best,
> > Authors

---

### Note · Authors · 2025-08-13

First of all, we thank all the Reviewers for their insightful and valuable comments. We are encouraged that the reviewers all give the high praise to our work including **novel perspective**   about ***perception-first*** RL (R:S8da, AEua, z4S9), **simple but effective approach** (R:S8da, AEua, z4S9, 8gLg), **extensive experiments and excellent performance** (R:S8da, AEua, z4S9, 8gLg), **deep and insightful analysis** (R:S8da, AEua, z4S9, 8gLg), and **good writing and high reproducibility** (R:S8da, AEua, z4S9).

Subsequently, during the rebuttal and discussion phase, we focused on addressing and elaborating on the following core concerns:

* **Theoretical explanation of the comparison between rule-based RL and SFT.** Regarding this point, we provided a comprehensive, three-pronged discussion (see response to **Q1, R:S8da** and **Q2, R:AEua**) on the advantages of rule-based RL in comparison to SFT.
* **Inclusion of more baseline and experimental results.** We have supplemented our work with additional experiments, incorporating the performance of *LLaVA-1.5* and *Qwen2/2.5-VL* as baselines (see response to **Q3, R:S8da**, **Q3, R:AEua**, and **Q4, R:z4S9**). Furthermore, we have evaluated our method on real-world benchmarks (see response to **Q5, R:AEua**), and have also conducted a more detailed generalization analysis (see response to **Q3, R:z4S9**).
* **Theoretical definition and clarification of the proposed "Perception Policy" and "Perceptual Perplexity".** We have added rigorous definitions and formal mathematical expressions for these two concepts. (see response to **Q2, R:S8da**, **Q1, R:z4S9**)
* **Elaboration on the task of scaling up perception.** We have provided a detailed explanation and exposition of the design rationale behind Perception-R1 (see follow-up response to **Q1, R:AEua**, and **Q1, R:8gLg**), highlighting its significant value in terms of the scalability of perception environment construction. In doing so, we propose a viable future direction for the next generation of multimodal reinforcement learning (see follow-up response to **R:8gLg**).

We have provided point-by-point responses to each reviewer's concerns. We are encouraged by the positive reception and believe this discussion has significantly strengthened the quality of our work on Perception-R1.

Finally, we thank the AC and reviewers for their invaluable time and trust that our clarifications will prove valuable in your forthcoming discussions and final decision.

---

### Decision · Program_Chairs · 2025-09-17

**Decision:**

Accept (poster)

**Comment:**

This paper proposes to improve multimodal LLMs in object detection, counting, and visual grounding through rule-based RL (with GRPO). The authors make several insightful findings, such as explicit chain-of-thought can hurt perception, and that the effectiveness of RL is correlated with the task's "perceptual perplexity."

The reviewers raised valid concerns regarding the conceptual rigor of terms like "perception policy," the lack of comparison to preference-based methods, and using a highly specialized base model. The authors responded by offering formal definitions for their key concepts, provided strong theoretical justifications for their choice of RL algorithm, and ran additional ablation studies. The new experiment applying their framework to Qwen2-VL-2B demonstrated that the performance gains are attributable to their method and not just the specialized base model.

The reviewers were largely satisfied with the authors' responses, with 3/4 recommending acceptance and raising their scores after the rebuttal period. The paper is well-executed, the results are significant, and it opens up a valuable new direction for grounding multimodal LLMs. The paper makes valuable contribution to the conference.